# Schematic memory components converge within angular gyrus during retrieval

Isabella C Wagner[1,2]*, Mariët van Buuren[1], Marijn CW Kroes[1,3,4], Tjerk P Gutteling[2], Marieke van der Linden[1], Richard G Morris[5], Guillén Fernández[1]

[1]Donders Institute for Brain, Cognition and Behaviour, Radboudumc, Nijmegen, The Netherlands; [2]Donders Institute for Brain, Cognition and Behaviour, Radboud University, Nijmegen, The Netherlands; [3]Center for Neural Science, New York University, New York, United States; [4]Department of Psychology, New York University, New York, United States; [5]Centre for Cognitive and Neural Systems, University of Edinburgh, Edinburgh, United Kingdom

*For correspondence: i.wagner@donders.ru.nl

**Abstract** Mental schemas form associative knowledge structures that can promote the encoding and consolidation of new and related information. Schemas are facilitated by a distributed system that stores components separately, presumably in the form of inter-connected neocortical representations. During retrieval, these components need to be recombined into one representation, but where exactly such recombination takes place is unclear. Thus, we asked where different schema components are neuronally represented and converge during retrieval. Subjects acquired and retrieved two well-controlled, rule-based schema structures during fMRI on consecutive days. Schema retrieval was associated with midline, medial-temporal, and parietal processing. We identified the multi-voxel representations of different schema components, which converged within the angular gyrus during retrieval. Critically, convergence only happened after 24-hour-consolidation and during a transfer test where schema material was applied to novel but related trials. Therefore, the angular gyrus appears to recombine consolidated schema components into one memory representation.

## Introduction

Associative knowledge structures in the form of so-called "mental schemas" (*Bartlett, 1932*) are built on the basis of several encounters with similar material. They may be applicable to a wide range of instances in which new information is integrated into established or newly established knowledge (*Ghosh and Gilboa, 2014*), and thereby promote encoding and subsequent consolidation (*Tse et al., 2007*, *2011*; *van Kesteren et al., 2010b*). This beneficial "schema effect" has been associated with hippocampal and medial prefrontal processing (*Tse et al., 2007*, *2011*; *Kumaran et al., 2009*; *van Kesteren et al., 2010a*, *2010b*, *2013*; *Dragoi and Tonegawa, 2013*; *McKenzie et al., 2014*), which is shifted towards a more neocortically centered system after consolidation (*Frankland and Bontempi, 2005*; *Takashima et al., 2006*; *Takehara-Nishiuchi and McNaughton, 2008*).

Despite the importance of schemas for learning, memory, and education, the current field is lacking a consistent definition. So far, attempts to operationalize schemas spanned an entire spectrum, ranging widely from simple, rule-like associations (if A-B, and B-C, then A-C; *Preston and Eichenbaum, 2013*), and more complex visuo-spatial layouts (*Tse et al., 2007*, *2011*; *van Buuren et al., 2014*), to pre-existing real-world knowledge (students remember new study material related to their own field better than material from other disciplines; *van Kesteren et al., 2014*). Considering this

**eLife digest** To make sense of the world around us, we constantly try to work out the relationship of new information to other things that we already know, and sort our knowledge into pre-existing mental frameworks, or "schemas". This makes learning new things that are related to a schema, as well as remembering this knowledge, easier. The process of making these mental connections is thought to involve an extensive brain network. Separate types of information are stored in different brain regions within this network, yet to link this information together, the brain must combine them into a single representation.

Wagner et al. have now investigated which brain regions are involved in recombining separate information. Human volunteers were trained to interpret the positions or colors of pairs of circles with different rules. The combination of these separate types of information formed a mental schema that could be used as a "weather forecast". The design of the experiment meant that measuring the brain activity of the volunteers during the task (using a technique called functional magnetic resonance imaging) allowed the brain regions involved in retrieving the different parts of such a schema to be distinguished.

Twenty-four hours later volunteers returned to use the mental schemas that they had learned to predict the weather. Retrieving which weather conditions the circle pairs represented activated a network of regions in the volunteers' brains. Further analysis revealed that some of these regions showed specific activity patterns in response to remembering information about only one element of the task (for example, only the rules or only the visual information). However, the different aspects of the task all appeared to be integrated by a brain region called the angular gyrus. This suggests that the angular gyrus is responsible for combining separate memory parts and pieces of information into a single representation. It is able to do so by connecting to brain regions that code for such specific aspects, although this only occurs 24 hours after the mental schemas have been established.

Future studies could investigate the result of damage to the angular gyrus: different pieces of information might not be combined, or could result in an incorrect memory during retrieval. Finally, since the angular gyrus has been related to a wealth of different mental processes, it remains a challenge for future research to "converge" these findings and to understand the underlying computations.

spectrum of complexity, it remains an empirical question whether there is a clear border between simple sets of rules and schemas and if so, where this border should be drawn (*Kroes and Fernandez, 2012*). Regardless of these various definitions, schema memories are thought to be facilitated by a distributed system that stores components as separate "units" (*Bartlett, 1932*; *Schacter et al., 1998*), or "features" (*van Kesteren et al., 2010a*), and that relies on inter-connected networks of neocortical representations (*Wang and Morris, 2010*). By the same token, this argues for the need to converge information in order to recombine associative schema components upon retrieval. Exactly where in the brain such recombination takes place is, however, still unclear.

The medial prefrontal cortex (MPFC) and hippocampus (HC), together with the parahippocampal cortex (PHC), posterior cingulate cortex (PCC), and angular gyrus (AG) have been identified as regions forming a network that is important for successful (episodic) memory retrieval (*Rugg and Vilberg, 2013*; *Watrous et al., 2013*; *King et al., 2015*). Especially the MPFC is involved in the retrieval of schemas (*Tse et al., 2011*; *Kroes and Fernandez, 2012*; *Ghosh et al., 2014*; *Richards et al., 2014*; *Warren et al., 2014*) and, during this process, establishes functional connections to posterior representation regions (*Marr, 1970*; *Frankland and Bontempi, 2005*; *van Kesteren et al., 2010a*). Furthermore, the AG seems well suited to support integrative retrieval (*Wagner et al., 2005*; *Gilmore et al., 2015*), since it has been discussed to guide the "binding", or recombination, of information (*Binder et al., 2009*; *Shimamura, 2011*; *Price et al., 2015*).

In the present study, we asked where different schema components are neuronally represented and where such representations converge into a comprehensive signature during retrieval. We followed several steps to test this question: First, as mental schemas are dependent on memory consolidation (*Tse et al., 2007*), we identified regions associated with this process. Second, we probed the functional coupling of these regions as they form a memory retrieval network. Third, and most

importantly, we identified the distributed representations of schema components and tested where such representations would converge. We defined schemas as sets of conceptual, rule-based associations (*Kumaran et al., 2009*), and reasoned that this approach would provide us with a well-controlled vehicle to establish the nature of schema-related retrieval in humans.

Subjects underwent fMRI during repeated, high-confident retrieval of two schemas (day 2) that were trained on a previous day (day 1; *Figure 1A*). These schemas were incorporated into a modified, deterministic weather prediction task (*Knowlton et al., 1994*; *Kumaran et al., 2009*) in which subjects had learned that colored circle pairs predicted specific but fictive weather outcomes ("sun", "rain"), depending on the location (spatial schema) or color (non-spatial schema) of one of the circles (*Figure 1B*; for a detailed description please see **Materials and methods, Material and task**). Crucially, our controlled design allowed us to independently capture the different schema components. During retrieval, visually presented circle pairs had to be combined with abstract rule-based information and could thus be used to predict specific trial outcomes. The combination of these different levels of information formed a simple schema. Therefore, the schema components consisted of (1) rule-based associations, and (2) low-level visual features of the task material (*Figure 1—figure supplement 1A*). Considering whole-brain characteristics rather than zooming into the functional properties of isolated regions, we employed a combination of activation, connectivity, and multi-voxel pattern analyses. We hypothesized that schema retrieval would primarily engage neocortical midline

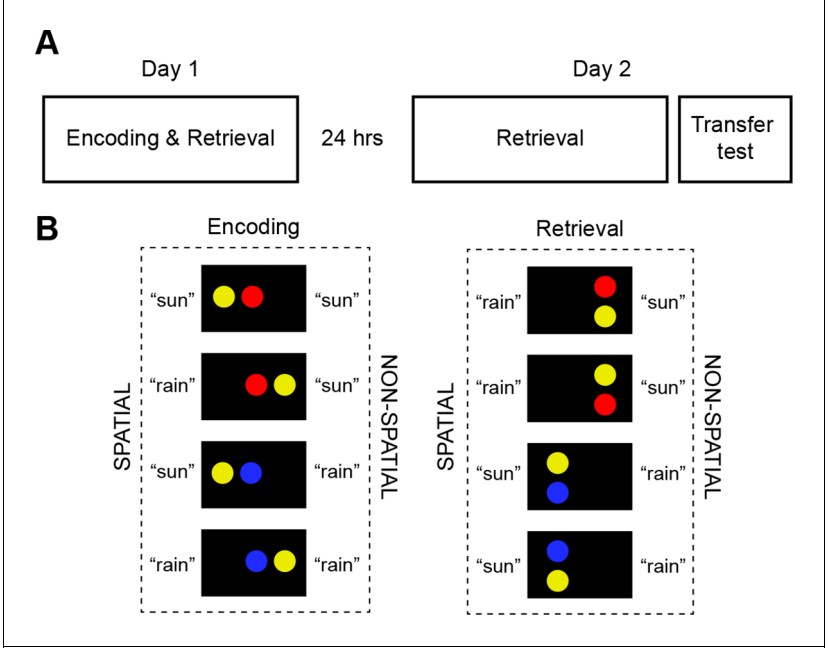

**Figure 1.** Study timeline and schema material. (**A**) Subjects underwent fMRI on two consecutive days, each containing 7 runs (day 1: encoding and retrieval; day 2: retrieval). A transfer test was completed at the end of day 2 and consisted of two runs (encoding and retrieval). (**B**) Stimulus material during encoding comprised four horizontal circle pairs. Spatial (position) or non-spatial (color) rule-based schemas were used to predict fictive "sun" or "rain" outcomes. Stimulus material during retrieval consisted of four vertical circle pairs (for a detailed description of the experiment, please see **Materials and methods, Material and task** and **Materials and methods, Procedure**). We used retrieval trials on day 2 to dissociate the multi-voxel patterns of schema components that consisted of rule-based associations and low-level visual features (*Figure 1—figure supplement 1*). *Figure 1—figure supplement 2* illustrates experimental trials during encoding, retrieval, and during the perceptual baseline condition.

The following figure supplements are available for Figure 1:

**Figure supplement 1.** Multi-voxel pattern analysis (MVPA).

**Figure supplement 2.** Experimental trials.

structures, such as the MPFC. Further, the MPFC should act as a convergence zone that recombines the different schema components into a unique schema memory during retrieval. Additionally, if such recombination goes beyond MPFC-centered processing, we expected retrieval-related schema representations to be held by the AG.

## Results

### Behavioral performance

Subjects acquired schemas across seven runs throughout day 1. These runs were structured in blocks of interleaved encoding and retrieval (**Materials and methods, Procedure** and **Materials and methods, Schema encoding**). Retrieval trials did not provide feedback and thus allowed us to estimate rule-based schema proficiency at steady time-points. Further, these trials required the application of schema knowledge to related information (vertical as opposed to horizontal arrangement of circle pairs, see *Figure 1B*). Investigating performance during schema retrieval, we found a significant three-way interaction of day (day 1, day 2) × run (1 to 7) × schema (spatial, non-spatial) ($F$ (3.8,63.7) = 3.3, $P$ = 0.017; interaction day × run: $F$(6,102) = 4.4, $P$ = 0.001; interaction day × schema: $F$(1,17) = 14.3, $P$ = 0.002; interaction run × schema: $F$(3.7,62.6) = 2.7, $P$ = 0.043; no main effect of day: $P$ = 0.062; no main effect of run: $P$ = 0.154; no main effect of schema: $P$ = 0.057). This interaction was followed-up by separate repeated measures ANOVAs for each day, with run and schema as within-subject factors. Only on day 1 we observed a significant interaction between runs and schemas ($F$(3,50.1) = 3.9, $P$ = 0.014; main effect of run: $F$(3.4,56.9) = 3.8, $P$ = 0.011; main effect of schema: $F$(1,17) = 7.7, $P$ = 0.013), which was caused by lower performance in the spatial as compared to non-spatial condition during the first run ($t$(21) = -3.2, $P$ = 0.005; *Figure 2A*, left). Throughout day 2, retrieval performance did not differ significantly between runs or conditions (no main effect of run: $P$ = 0.334; no main effect of schema: $P$ = 0.666; no run × schema interaction: $P$ = 0.761; *Figure 2A*, right).

Similarly, reaction times (RTs) and retrieval confidence between schema conditions only differed during the first run of day 1 (*Figure 2B, C*, left). Here, we found longer RTs ($t$(21) = 2.47, $P$ = 0.022) and lower retrieval confidence ($t$(21) = -4.2, $P$ < 0.0005) for the spatial as compared to the non-spatial schema. On day 2, we did not find any significant differences between runs or schema conditions in terms of RTs (no main effect of run: $P$ = 0.718; no main effect of schema: $P$ = 0.749; no run × schema interaction: $P$ = 0.849; *Figure 2B*, right), or retrieval confidence (no main effect of run: $P$ = 0.187; no main effect of schema: $P$ = 0.397; no run × schema interaction: $P$ = 0.549; *Figure 2C*, right; for details see **Materials and methods, Schema retrieval: reaction times** and **Materials and methods, Schema retrieval: confidence**). In summary, subjects rapidly learned to apply both rule-based schemas. Retrieval performance was stable at the end of day 1 and throughout day 2, and did not differ between the conditions.

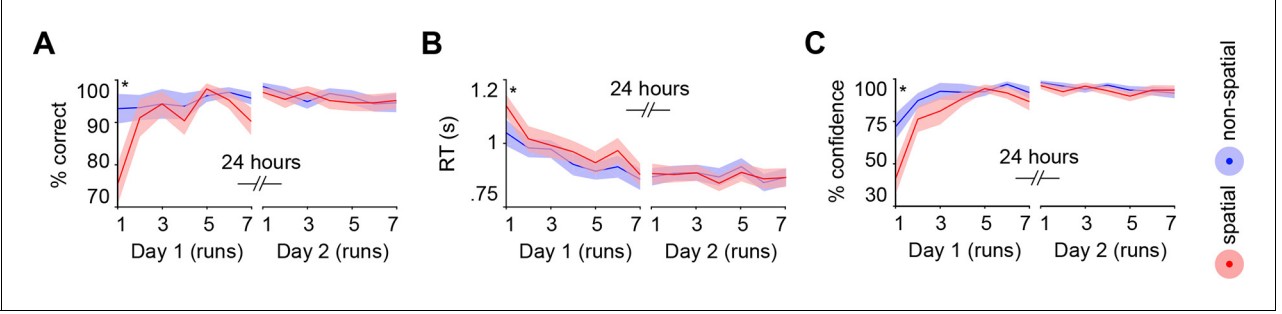

**Figure 2.** Behavioral performance during schema retrieval. (**A**) Data represents the % of correct responses, (**B**) the average reaction time (s), and (**C**) the % of high-confident ratings (i.e. "sure"-responses). Shaded error bars denote ± standard error of the mean (s.e.m.). * marks a significant ($P$ < 0.05) difference between the schema conditions within the first run of day 1.

## Schema consolidation

Across both days, schema retrieval was associated with increased blood oxygen-level dependent (BOLD) responses in bilateral lingual gyrus, superior occipital gyrus, cuneus, left supplemental motor area, and right parahippocampal cortex (day 1 & day 2; *Figure 3A*; *Table 1*, upper part). Since consolidation is considered a prerequisite for mental schemas (*Tse et al., 2007*), we next performed a contrast between days. We asked whether retrieval would yield increased engagement of neocortical midline regions after a delay of 24 hours. Additionally, we controlled for differences in schema performance, confidence, and RTs by performing a specific contrast between runs that were similar with regard to these aspects (i.e. the last three runs of day 1 with the first run of day 2; approx. 48 vs. 32 trials, respectively). Behaviorally, subjects were able to retrieve and confidently apply both schemas (*Figure 2*, left and right). Schema retrieval performance and confidence did not differ significantly between runs or schema conditions (retrieval performance: no main effect of run: $P = 0.103$; no main effect of schema: $P = 0.173$; no run $\times$ schema interaction: $P = 0.437$; confidence: no main effect of run: $P = 0.261$; no main effect of schema: $P = 0.16$; no run $\times$ schema interaction: $P = 0.427$). Further, there was no significant difference in RTs (no main effect of run: $P = 0.09$; no main effect of schema: $P = 0.355$; no run $\times$ schema interaction: $P = 0.547$; *Figure 2*, middle). This specific comparison yielded increased activation in bilateral lingual gyrus, superior occipital gyrus, cuneus, and left supplemental motor area on day 1 relative to day 2 (day 1 > day 2; *Figure 3B*; *Table 1*, middle part). After initial consolidation, activation was increased in PCC, precuneus, and MPFC, as well as in a set of right lateralized regions including the supramarginal gyrus, middle temporal gyrus, and inferior temporal gyrus (day 2 > day 1; *Figure 3C*; *Table 1*, lower part). Conclusively, we found stronger retrieval-related activation within MPFC, PCC, and higher-level sensory regions after a 24-hour-delay.

## Schema retrieval networks: MPFC and PCC

So far, we identified stronger retrieval-related activation within MPFC and PCC after a 24-hour-delay (day 2 > day 1; **Results, Schema consolidation** and *Figure 3C*); and here we used this contrast to derive seed regions for our following connectivity analyses. We applied Psychophysiological Interaction analysis (PPI; **Materials and methods, Connectivity analysis**) to identify the connectivity profiles of the two rule-based schemas during retrieval as compared to the perceptual baseline (*Figure 1— figure supplement 2C*) on day 2, and placed seeds within MPFC (x = -2, y = 35, z = -2) and PCC (x = 2, y = -45, z = 22).

First, we investigated functional coupling of the MPFC (*Figure 4A*; *Table 2*, upper part): During retrieval of the spatial schema, the MPFC was more strongly coupled to surrounding medial prefrontal regions, HC and PHC, PCC, precuneus, and left AG. For non-spatial retrieval, the MPFC showed enhanced coupling with its locally surrounding regions. However, lowering the statistical threshold ($P < 0.005$, uncorrected) revealed comparable results for both conditions. Further, there were no

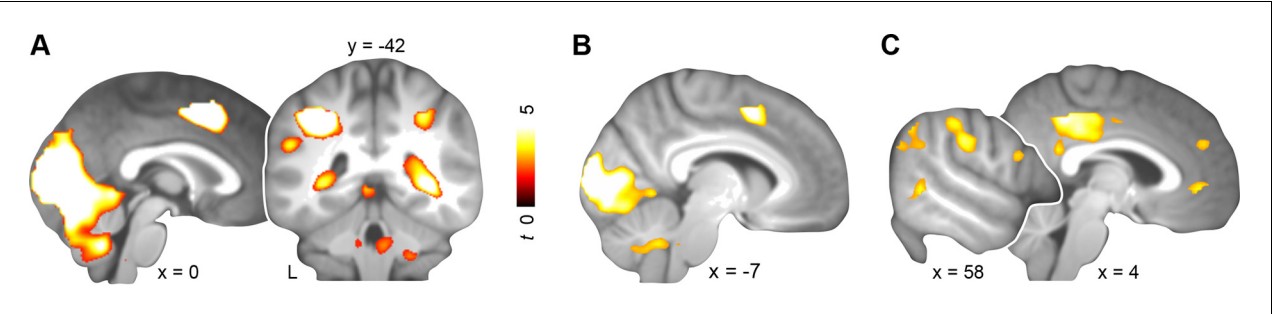

**Figure 3.** Activation during schema retrieval. (A) Increased BOLD responses during rule-based schema retrieval across both days (schema retrieval > perceptual baseline), (B) during rule-based schema retrieval on day 1 (day 1 > day 2), and (C) after an initial consolidation of 24 hours (day 2 > day 1). Contrasts B and C include runs 5 to 7 from day 1, and the first run from day 2. For display purposes, results were resliced to a voxel dimension of 0.5 mm isotropic and are shown at $P < 0.001$, uncorrected. Significant clusters are noted in *Table 1*. Results are superimposed onto the average structural scan derived from all subjects. L–left.

**Table 1.** Activation during schema retrieval.

| Brain region | MNI | | | Z value | Cluster size |
|---|---|---|---|---|---|
| | x | y | z | | |
| **Day 1 & day 2** | | | | | |
| L superior frontal gyrus | -5 | 5 | 52 | | 766 |
| L superior parietal gyrus | -22 | -60 | 40 | | 9440 |
| L middle frontal gyrus | -28 | -5 | 45 | | 938 |
| R middle frontal gyrus | 32 | -2 | 48 | | 260 |
| L insular cortex | -32 | 20 | 8 | 5.53 | 126 |
| **Day 1 > day 2** | | | | | |
| L cuneus | -2 | -95 | 10 | | 2814 |
| L superior frontal gyrus | -5 | 5 | 52 | 6.14 | 165 |
| Cerebellum | -35 | -50 | -32 | 4.37 | 161 |
| **Day 2 > day 1** | | | | | |
| L cingulate gyrus | 0 | -40 | 42 | 5.38 | 593 |
| R supramarginal gyrus | 55 | -22 | 30 | 4.32 | 156 |
| R superior frontal gyrus | 5 | -45 | -2 | 3.93 | 95 |
| R middle temporal gyrus | 58 | -50 | 0 | 3.85 | 106 |

Clusters that showed significant BOLD increases during retrieval of rule-based schema memories across days, before, and after a 24-hour-delay. Bold font indicates contrasts. Retrieval was compared to the perceptual baseline. MNI coordinates represent the location of peak voxels. We report the local maximum of each cluster. Effects were tested for significance using cluster-inference with a cluster-defining threshold of $P < 0.001$ and a cluster-probability of $P < 0.05$ family-wise error (FWE) corrected for multiple comparisons (critical cluster size: 86). L – left, R – right.

significant connectivity differences between spatial and non-spatial schema retrieval (tested with a paired-sample $t$-test).

Second, we turned to the seed region within the PCC (*Figure 4B*; *Table 2*, lower part): Spatial schema retrieval was associated with enhanced functional coupling between the PCC and surrounding posterior midline regions, such as precuneus. The MPFC, HC, PHC, fusiform gyrus, and superior temporal gyrus also showed enhanced coupling with the PCC, along with the left AG. A similar network emerged during non-spatial schema retrieval. Again, there were no significant connectivity differences between the two conditions (tested with a paired-sample $t$-test).

To sum up, while the MPFC mainly showed increased neocortical coupling during spatial schema retrieval, the PCC was connected to an extensive network of regions during retrieval of both schema conditions (*Figure 4B*; *Table 2*, lower part). This network consistently involved MTL, MPFC, PCC, and left AG and constitutes a set of brain regions that was previously reported to underlie successful memory retrieval (*Rugg and Vilberg, 2013*; *Watrous et al., 2013*; *King et al., 2015*).

## Multi-voxel representations of schema components

Schemas are thought to be facilitated by a distributed system that stores memory components as separate representational units (*Bartlett, 1932*; *Schacter et al., 1998*). Here, we asked where such components are neuronally represented. Crucially, our design allowed us to individually capture schema components, defined as (1) rule-based associations, and (2) low-level visual features. We employed multi-voxel pattern analysis (MVPA) in combination with a whole-brain searchlight procedure on day 2 (**Materials and methods, Multi-voxel pattern analysis**), and separated schema components by discriminating (1) the schema conditions (while collapsing across visual features), and (2) the visual features (while collapsing across schema conditions; *Figure 1—figure supplement 1*).

First, we identified voxel patterns that discriminated between schema conditions (spatial vs. non-spatial), and that served as a marker for representations of rule-based associations. By keeping visual

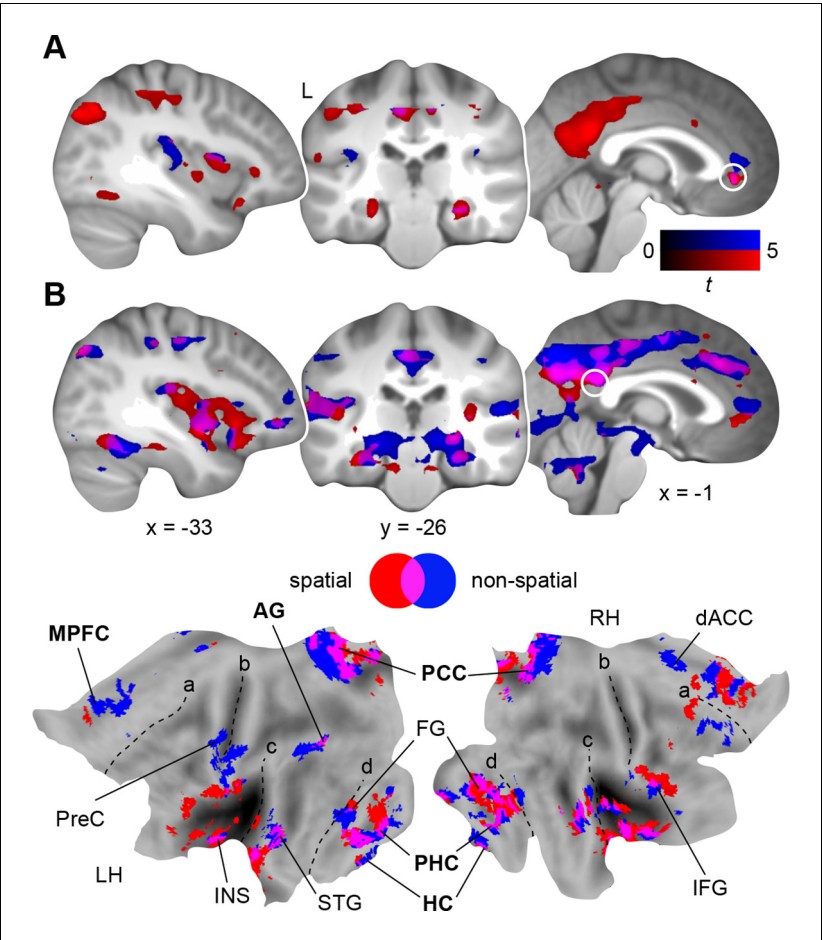

**Figure 4.** Schema retrieval networks: MPFC and PCC. (**A**) MPFC seed (x = -2, y = 35, z = -2; based on the contrast day 2 > day 1, *Figure 3C*; here marked in white). (**B**) PCC seed (x = 2, y = -45, z = 22; based on the same contrast; here marked in white). General retrieval effects are shown in purple (schema retrieval > perceptual baseline). For display purposes, connectivity maps were resliced to a voxel dimension of 0.5 mm isotropic and are shown at $P < 0.001$, uncorrected. Significant clusters are noted in *Table 2*. L – right. Additionally, connectivity results (PCC seed) are projected onto a surface-based flatmap. Relevant structures are labeled: AG, angular gyrus; dACC, dorsal anterior cingulate cortex; FG, fusiform gyrus; HC, hippocampus; IFG, inferior frontal gyrus; INS, insula; MPFC, medial prefrontal cortex; PCC, posterior cingulate cortex; PHC, parahippocampal cortex; PreC, precentral gyrus; STG, superior temporal gyrus. Regions of the retrieval network are highlighted in bold font. Dashed lines are inserted to aid orientation: a, border between medial and lateral prefrontal cortices; b, central sulcus; c, superior temporal gyrus; d, border between ventromedial and -lateral temporal cortices. LH – left hemisphere, RH – right hemisphere.

input between both conditions constant, we considered multi-voxel schema patterns that go beyond any visual representation of the different circle pairs (for example, a yellow and blue circle on the right half of the screen predict "sun" when applying the spatial schema, but the same circle pair predicts "rain" when applying the non-spatial schema; *Figure 1B*). Rule-based associations were represented in the right ventrolateral prefrontal cortex, left middle occipital gyrus, and left AG (*Figure 5A*; *Table 3*, upper part).

To determine brain regions that solely represent the low-level visual features of both schemas we next trained a classifier that dissociated the different circle pairs (circle pairs 1 and 2 vs. circle pairs 3 and 4, cancelling out the respective rule-based associations; *Figure 1—figure supplement 1B*). This allowed us to target the sum of visual features that formed the necessary basis to successfully apply one of the two schemas (namely color and position). In line with our expectation to identify discrimination performance primarily in the visual system, we revealed that low-level visual features were

**Table 2.** Schema retrieval networks: MPFC and PCC.

| Brain region | MNI | | | Z value | Cluster size |
|---|---|---|---|---|---|
| | x | y | z | | |
| Seed MPFC, spatial > perceptual baseline | | | | | |
| R superior frontal gyrus | 10 | 58 | 5 | 5.11 | 362 |
| L angular gyrus | -45 | -72 | 35 | 4.84 | 206 |
| R parahippocampal gyrus | 28 | -35 | -10 | 4.69 | 126 |
| L precuneus | -10 | -60 | 20 | 4.66 | 1008 |
| L precentral gyrus | -52 | -12 | 45 | 4.11 | 96 |
| L parahippocampal gyrus | -28 | -38 | -10 | 3.82 | 141 |
| Seed MPFC, non-spatial > perceptual baseline | | | | | |
| L cingulate gyrus | -5 | 38 | 8 | 4.26 | 314 |
| Seed PCC, spatial > perceptual baseline | | | | | |
| L cingulate gyrus | -10 | -45 | 8 | 5.07 | 1746 |
| R cingulate gyrus | 12 | 30 | 20 | 4.68 | 685 |
| L precentral gyrus | -55 | 8 | 2 | 4.64 | 1002 |
| R insular cortex | 35 | -22 | 8 | 4.62 | 120 |
| L parahippocampal gyrus | -32 | -35 | -15 | 4.49 | 553 |
| R inferior frontal gyrus | 52 | 18 | 12 | 4.44 | 279 |
| L angular gyrus | -48 | 70 | 40 | 4.42 | 106 |
| R superior temporal gyrus | 45 | -2 | -12 | 4.30 | 663 |
| Cerebellum | 12 | -72 | -28 | 4.23 | 142 |
| Cerebellum | -15 | -58 | -35 | 4.00 | 164 |
| Seed PCC, non-spatial > perceptual baseline | | | | | |
| L precuneus | -2 | -65 | 30 | 5.49 | 3597 |
| L precentral gyrus | -55 | -8 | 45 | 4.89 | 680 |
| R middle temporal gyrus | 65 | -18 | -8 | 4.58 | 716 |
| R superior frontal gyrus | 12 | 25 | 28 | 4.27 | 587 |
| L angular gyrus | -48 | -70 | 38 | 4.19 | 184 |
| Cerebellum | -20 | -68 | -28 | 4.03 | 282 |
| R middle frontal gyrus | 18 | 65 | 12 | 3.81 | 156 |

Clusters that showed a significant increase in connectivity during schema retrieval: MPFC (x = -2, y = 35, z = -2) and PCC (x = 2, y = -45, z = 22). Bold font indicates contrasts. Retrieval was compared to the perceptual baseline. MNI coordinates represent the location of peak voxels. We report the local maximum of each cluster. Effects were tested for significance using cluster-inference with a cluster-defining threshold of $P < 0.001$ and a cluster-probability of $P < 0.05$ family-wise error (FWE) corrected for multiple comparisons (critical cluster sizes; MPFC seed: spatial, 89 voxels; non-spatial, 95 voxels; PCC seed: 89 voxels for both conditions). L – left, R – right.

represented in the lingual gyrus, fusiform gyrus, middle occipital gyrus, cuneus, and the AG (*Figure 5A*; *Table 3*, middle part).

Since a distributed schema memory system is expected to rely on inter-connected networks of neocortical representations (*Wang and Morris, 2010*), we expected the convergence of both schema components within the retrieval network (see above, *Figure 4*). These components consisted of rule-based associations and low-level visual features – whereby a combination of both was necessary to solve a given trial successfully. Most importantly, we hypothesized a functional role of the MPFC or AG in retrieval-related schema integration. As can be seen from *Figure 5A* (magnified cutouts), both levels of schema components overlapped within the left AG. We did not find retrieval-related convergence of schema components within the MPFC.

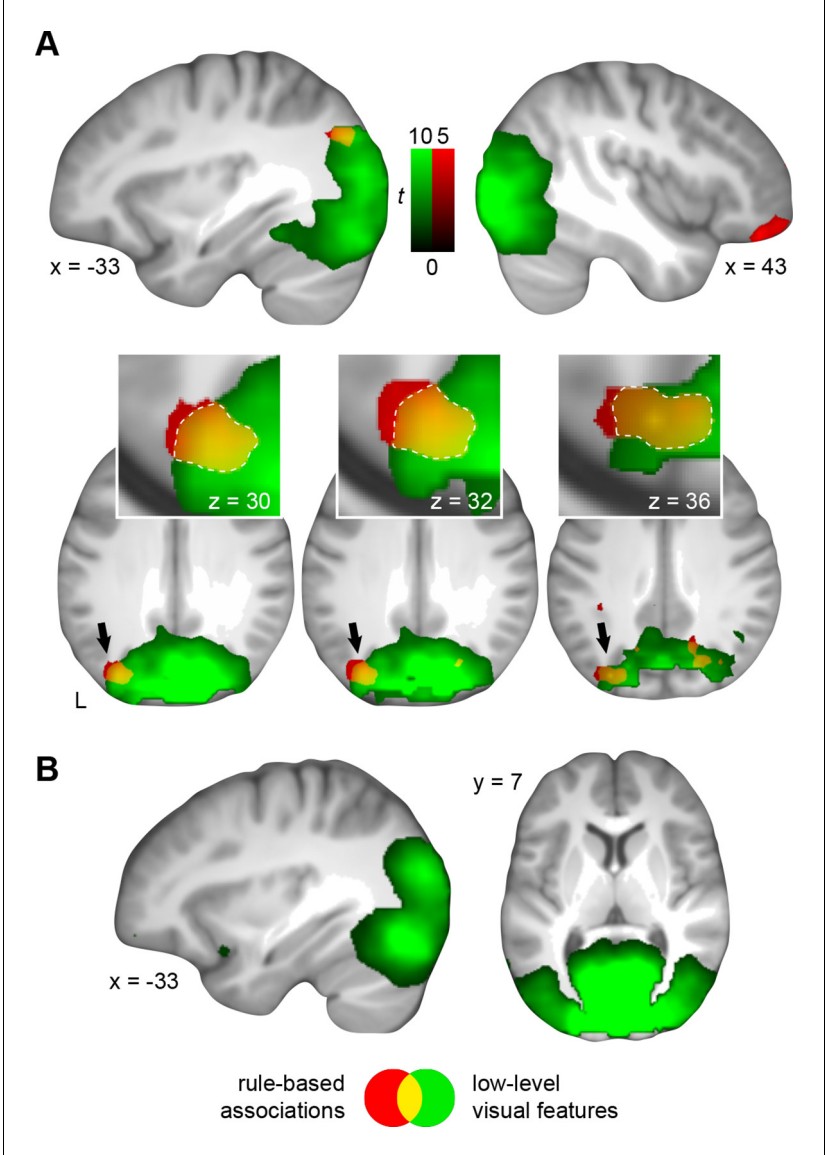

**Figure 5.** Multi-voxel representations of schema components. (**A**) The searchlight MVPA revealed distributed representations of both schema components (rule-based associations, low-level visual features). These representational levels converged within the AG (yellow). Three horizontal slices are shown as cut-outs and are magnified to appreciate the overlap (*Table 3*). (**B**) Only the multi-voxel patterns of low-level visual features were shared between day 1 and day 2. For display purposes, all maps were resliced to a voxel dimension of 0.5 mm isotropic and are shown at *P* < 0.001, uncorrected. Significant clusters are noted in *Table 3*. L – left.

Next, we reasoned that if schema components converged in the AG only after consolidation, their multi-voxel representations should only partly generalize from day 2 to day 1. That is, training a classifier on day 2 and testing it on data from day 1 (*Figure 1—figure supplement 1C*) should not yield discrimination performance above chance level for rule-based associations. Although low-level visual features were connected to higher-level information and were thus also regarded as a schema component, circle pairs were visually presented on the screen on both days. Therefore, we expected significant discrimination performance for low-level visual features in occipital cortex regions.

As expected, representations of low-level visual features generalized from day 2 to day 1, indicated through significant discrimination performance in occipital cortex (*Figure 5B*; *Table 3*, lower part). For rule-based associations, none of the runs on day 1 showed discrimination performance significantly above chance level, implying that the multi-voxel representations of the spatial and non-

**Table 3.** Multi-voxel representations of schema components.

| Brain region | MNI | | | Z value | Cluster size |
|---|---|---|---|---|---|
| | x | y | z | | |
| **MVPA day 2, rule-based associations** | | | | | |
| R lateral orbitofrontal gyrus | 42 | 42 | -18 | 4.14 | 75 |
| L middle occipital gyrus | -30 | -75 | 32 | 4.13 | 102 |
| L angular gyrus | -38 | -70 | 32 | 3.73 | |
| **MVPA day 2, low-level visual features** | | | | | |
| L cuneus | 0 | -82 | 8 | | 16630 |
| **MVPA day 1, low-level visual features** | | | | | |
| R lingual gyrus | 2 | -78 | -2 | | 15599 |

Clusters that significantly discriminated schema component representations (rule-based associations, low-level visual features). Bold font indicates the type of MVPA analysis (day 1, training the classifier on day 2 and testing it on day 1; day 2, training the classifier on day 2 and testing it on day 2 using cross-validation; **Materials and methods, Multi-voxel pattern analysis**). MNI coordinates represent the location of peak voxels. We report the first two local maxima (> 8 mm apart) within each cluster (rule-based associations), and the local maximum for the low-level visual feature MVPAs. Effects were tested for significance using cluster-inference with a cluster-defining threshold of $P < 0.001$ and a cluster-probability of $P < 0.05$ family-wise error (FWE) corrected for multiple comparisons (critical cluster sizes: day 2, rule-based associations, 74 voxels; day 2, low-level visual features, 72 voxels; day 1, low-level visual features, 70 voxels). L – left, R – right.

spatial schema conditions were not shared between days. However, this does not preclude the involvement of the AG in schema retrieval prior to 24-hour-consolidation, but may be caused by representational differences between the days. Therefore, we additionally trained and tested a classifier on data from day 1. Again, we did not find representations of rule-based associations within the AG and thus no retrieval-related convergence of schema components on day 1 (**Materials and methods, Complementary analysis: AG involvement in schema retrieval on day 1),** suggesting that the left AG recombines schema components only after a 24-hour-delay.

## Schema convergence networks

Using MVPA, we demonstrated the retrieval-related convergence of schema components within the left AG after a 24-hour-delay. To support this convergence, the AG should show increased functional coupling with regions that separately represent the different components. We tested this assumption using PPI (**Materials and methods, Connectivity analysis**) and created a mask of the overlap between both schema components during retrieval on day 2 (*Figure 6*, right middle panel, marked in white; **Results, Multi-voxel representations of schema components**; *Figure 5A*). This mask was used as a seed region. Spatial schema retrieval (compared to the perceptual baseline) was associated with enhanced functional coupling between the left AG and its locally surrounding lateral parietal cortex. Further, we observed increased connectivity with the HC, PHC, MPFC, PCC, and fusiform gyrus (similar effects were observed for non-spatial schema retrieval; *Figure 6*, *Table 4*). Connectivity profiles between the two conditions did not differ significantly (tested with a paired-sample *t*-test). The fusiform finding appears particularly relevant, since the fusiform gyrus was shown to represent the low-level visual features of the schema material (see above, and *Figure 5A*). This corroborates our assumption that retrieval-related convergence within the AG is accomplished via increased functional connectivity among a distributed set of regions that each hold specific schema components. Moreover, the left AG was coupled to the retrieval network we identified earlier (see above, and *Figure 4*). The consistency of our PPI and MVPA results is further demonstrated in *Figure 7*.

## Transfer test: new schema encoding and retrieval

Schemas provide knowledge structures that help new but related information to be integrated more rapidly (*Tse et al., 2007*; *van Kesteren et al., 2014*). Therefore, our schema material should

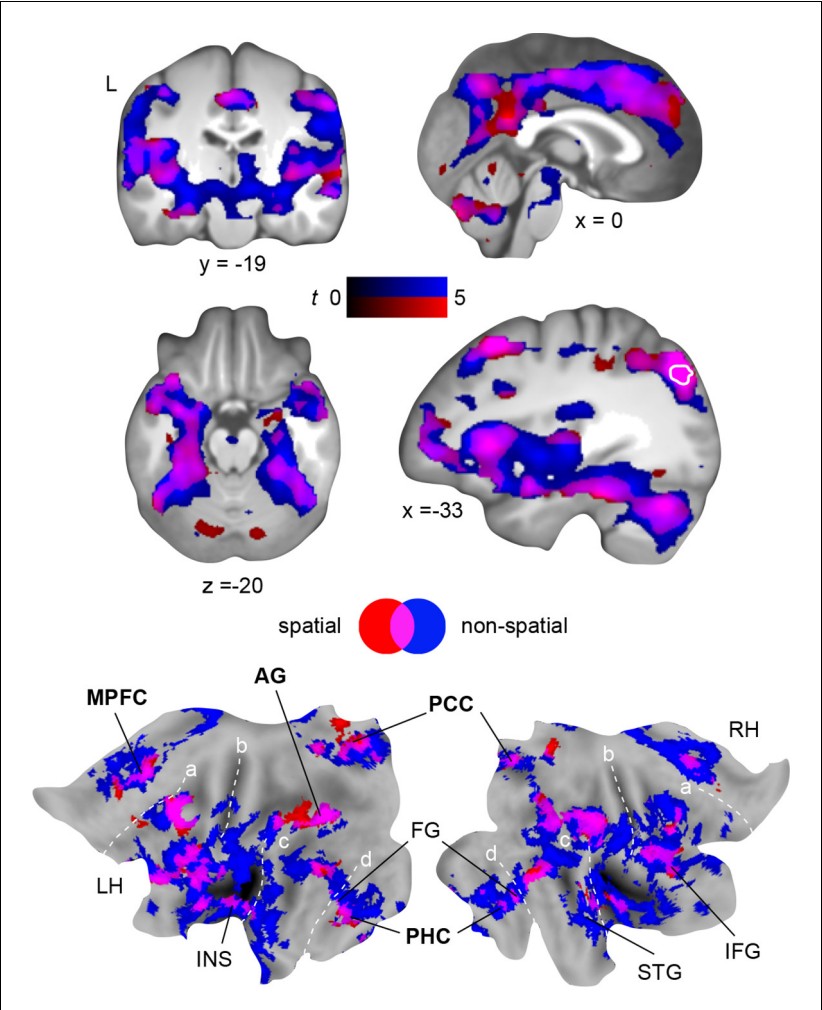

**Figure 6.** Schema convergence networks. Functional connectivity of the left AG seed (defined as cluster of overlapping schema components; based on our MVPA result, *Figure 5A*, here marked in white) during schema retrieval (compared to the perceptual baseline). General retrieval effects are shown in purple. For display purposes, maps were resliced to a voxel dimension of 0.5 mm isotropic and are shown at $P < 0.001$, uncorrected. L – left. Significant clusters are noted in *Table 4*. Additionally, connectivity results are projected onto a surface-based flatmap. Relevant structures are labeled: AG, angular gyrus; FG, fusiform gyrus; IFG, inferior frontal gyrus; INS, insula; MPFC, medial prefrontal cortex; PCC, posterior cingulate cortex; PHC, parahippocampal cortex; STG, superior temporal gyrus. Regions of the retrieval network are highlighted in bold font. Dashed lines are inserted to aid orientation: a, border between medial and lateral prefrontal cortices; b, central sulcus; c, superior temporal gyrus; d, border between ventromedial and -lateral temporal cortices. LH – left hemisphere, RH – right hemisphere.

facilitate transfer to related task material. We tested this assumption during a transfer test at the end of day 2 (*Figure 1A*; **Materials and methods, Procedure**). Here, the stimulus set was changed into circle pairs with different colors while keeping the same pair-wise arrangement (**Materials and methods, Material and task**). By changing the color of the stimulus set, the transfer test only required transfer of the non-spatial schema condition. This allowed us to match the difficulty between old and new non-spatial rule-based associations while a change in position would have lead to an increase in difficulty for the spatial schema condition.

Performance during encoding trials was significantly lower in the first run of the transfer test (main effect of run: $F(1,22) = 12.2$, $P = 0.002$; *Figure 8A*, left). Here, subjects performed worse in generalizing the non-spatial schema (main effect of schema: $F(1,22) = 6.2$, $P = 0.021$; interaction run

**Table 4.** Schema convergence networks.

| Brain region | MNI x | y | z | Z value | Cluster size |
|---|---|---|---|---|---|
| **Spatial > perceptual baseline** | | | | | |
| L middle occipital gyrus | -30 | -80 | 32 | 5.90 | 553 |
| R middle frontal gyrus | 42 | 22 | 45 | 5.37 | 182 |
| L middle frontal gyrus | -32 | 18 | 52 | 5.19 | 380 |
| Cerebellum | 15 | -75 | -28 | 4.96 | 1553 |
| R inferior temporal gyrus | 55 | -58 | -12 | 4.94 | 259 |
| L inferior frontal gyrus | -40 | 20 | 2 | 4.61 | 922 |
| R angular gyurs | 42 | -65 | 50 | 4.56 | 701 |
| L fusiform gyrus | -32 | -35 | -25 | 4.50 | 352 |
| L middle frontal gyrus | -18 | 45 | 28 | 4.50 | 509 |
| L superior parietal gyrus | -15 | -60 | 18 | 4.39 | 261 |
| R superior parietal gyrus | 20 | -55 | 18 | 4.19 | 109 |
| R precuneus | 8 | -55 | 40 | 4.12 | 152 |
| R insular cortex | 38 | -8 | 0 | 4.08 | 95 |
| R superior temporal gyrus | 65 | -18 | 2 | 4.03 | 107 |
| R inferior frontal gyrus | 52 | 35 | 22 | 3.95 | 456 |
| **Non-spatial > perceptual baseline** | | | | | |
| L superior frontal gyrus | 0 | -5 | 48 | 5.87 | 22582 |
| L inferior frontal gyrus | -38 | 15 | 5 | 5.77 | |
| Cerebellum | -15 | -75 | -38 | 5.76 | |
| L angular gyrus | -32 | -78 | 42 | 5.63 | |
| R fusiform gyrus | 48 | -55 | -22 | 5.54 | |
| L inferior frontal gyrus | -35 | 28 | 2 | 5.52 | |
| R middle temporal gyrus | 62 | -12 | -12 | 5.43 | |
| R fusiform gyrus | 45 | -45 | -22 | 5.42 | |
| L middle frontal gyrus | -38 | 12 | 52 | 5.42 | |
| R superior frontal gyrus | 2 | 28 | 52 | 5.42 | |
| L superior temporal gyrus | -52 | -5 | -8 | 5.40 | |
| R superior frontal gyrus | 8 | 8 | 52 | 5.32 | |
| Cerebellum | -25 | -60 | -35 | 5.32 | |
| L middle temporal gyrus | -62 | -58 | 2 | 5.31 | |
| R superior frontal gyrus | 8 | 28 | 40 | 5.30 | |

Clusters that showed a significant increase in AG connectivity during schema retrieval. Retrieval was compared to the perceptual baseline. The seed was defined as overlap between schema components, as determined with MVPA (**Figure 5A**). Bold font indicates contrasts. MNI coordinates represent the location of peak voxels. We report the local maximum of each cluster. For the non-spatial schema condition we report the first 15 local maxima (> 8 mm apart). Effects were tested for significance using cluster-inference with a cluster-defining threshold of $P < 0.001$ and a cluster-probability of $P < 0.05$ family-wise error (FWE) corrected for multiple comparisons (critical cluster sizes: spatial, 88 voxels; non-spatial, 83 voxels). L – left, R – right.

$\times$ schema: $F(1,22) = 5$, $P = 0.036$; $t(22) = 2.8$, $P = 0.01$), but had already adapted schemas to the new stimulus set during run 2 ($P = 0.803$). RTs did not show any differences between runs ($P = 0.681$), schemas ($P = 0.5$), or any run $\times$ schema interactions ($P = 0.477$; **Figure 8A**, right).

During retrieval, subjects performed significantly worse in applying the non-spatial schema (no main effect of run: $P = 0.441$; main effect of schema: $F(1,22) = 5.9$, $P = 0.023$; no run $\times$ schema

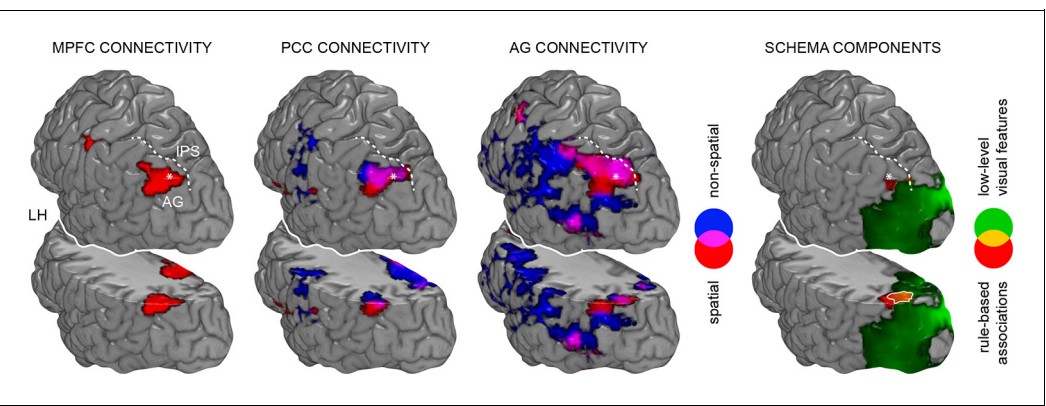

**Figure 7.** Spatial relationship between schema retrieval networks and schema component representations. Results from connectivity analyses (seeds MPFC, PCC, AG), and MVPA are shown as a 3D rendering. During schema retrieval, MPFC and PCC were functionally connected with the same AG region (left part). Furthermore, MVPA revealed distributed representations of different schema components that converged within the left AG during retrieval (right and surrounded in white). To aid orientation, dashed lines schematically indicate the intraparietal sulcus (IPS). Asterisks mark identical locations within the AG across the different methodological approaches. Additionally, we show a horizontal cut at the level of the AG to demonstrate sub-surface effects. LH – left hemisphere.

interaction: $P = 0.1$; *Figure 8B*, left), and were less confident (no main effect of run: $P = 0.17$; main effect of schema: $F(1,22) = 5.2$, $P = 0.033$; no run × schema interaction: $P = 0.105$; *Figure 8B*, right). However, during the final run of the transfer test, correct responses were delivered faster for the non-spatial schema ($t(22) = 5.1$, $P < 0.0005$; no main effect of run: $P = 0.44$; main effect of schema: $F(1,22) = 12.2$, $P < 0.005$; run × schema interaction: $F(1,22) = 5.9$, $P < 0.05$; *Figure 8B*, middle).

## Transfer test: comparison to initial schema acquisition

To investigate whether non-spatial schema knowledge was transferred from initial schema acquisition to new learning, we started out by comparing non-spatial schema performance, RTs, and retrieval confidence between the initial runs of day 1 and the transfer test.

Performance during schema encoding did not differ between the study phases ($P = 0.894$), but subjects responded significantly faster during the transfer test as compared to day 1 ($t(21) = 5.7$, $P < 0.0005$). Similarly, subjects responded faster when retrieving non-spatial schema material during the transfer test ($t(21) = 3.1$, $P = 0.006$), but retrieval performance and confidence did not differ significantly (retrieval performance: $P = 0.312$; retrieval confidence: $P = 0.244$). In conclusion, subjects responded faster during schema encoding and retrieval in the transfer test as compared to the initial run on day 1. We take this as indirect evidence that subjects applied schema knowledge to solve novel but related material.

## Transfer test: multi-voxel representations of schema components

In our final analysis, we tested the convergence of schema component representations during the transfer test. This analysis was grounded on the assumption that subjects would employ stable, consolidated schema knowledge to solve new task material (as suggested by our behavioral results above). If this was the case, the converging signatures of schema components should be similar between day 2 (prior to the transfer test) and the transfer test. Thus, training a classifier on data from day 2 and testing it on neural data from the transfer test (*Figure 1—figure supplement 1D*) should yield representations of rule-based associations and low-level visual features within the AG. MVPA was performed as described previously (**Materials and methods, Multi-voxel pattern analysis**).

In line with our prediction, and in contrast to day 1 (see above), rule-based associations were represented within the left middle occipital gyrus and AG (*Figure 8C*; *Table 5*, upper part). As on day 2, multi-voxel representations of low-level visual features were mainly found in occipital regions and AG (*Figure 8C*; *Table 5*, lower part). Most importantly, both levels of information converged in the

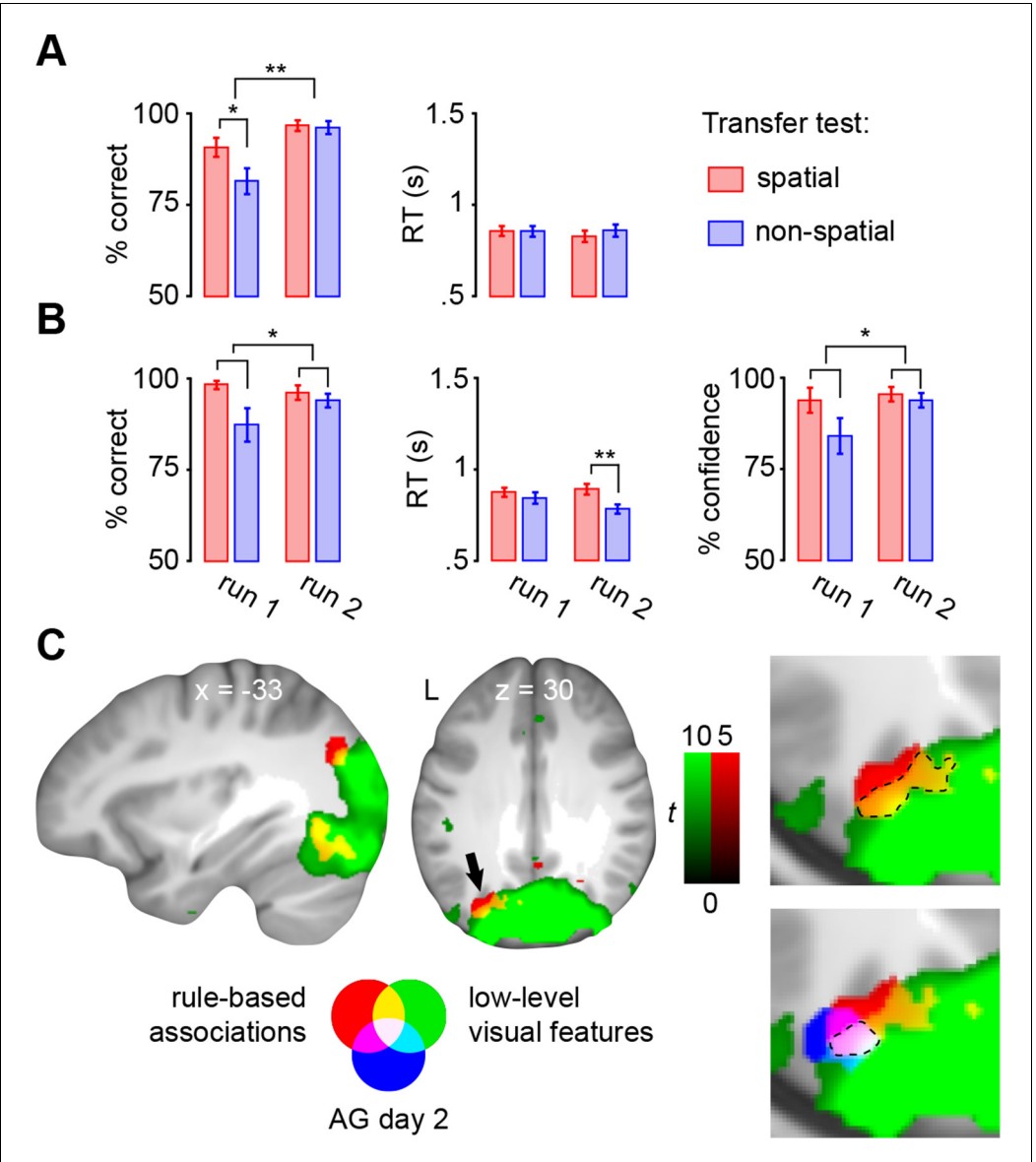

**Figure 8.** Transfer test. (**A**) Schema encoding: left, % of correct responses; right, average reaction time (s). (**B**) Schema retrieval: left, % of correct responses; middle, average reaction time (s); right, % of high-confident ratings (i.e. "sure"-responses). Error bars denote ± s.e.m. * marks significance at $P < 0.05$, ** marks significance at $P < 0.001$. (**C**) Multi-voxel patterns of rule-based associations and low-level visual features were shared between day 2 and the transfer test. Magnified cut-outs of the horizontal slice are provided to appreciate the overlap between schema components. The AG cluster showing schema convergence during day 2 is depicted in blue (**Figure 5A**). For display purposes, all maps were resliced to a voxel dimension of 0.5 mm isotropic and are shown at $P < 0.001$, uncorrected. Significant clusters are noted in **Table 5**. L – left.

left AG, and the precise location of convergence overlapped with our previous result (**Figure 8C**, marked in blue; and see **Figure 5A**). Therefore, neural signatures of schema components were similar between day 2 and the transfer test, suggesting that subjects applied schema material to new and related information. Furthermore, this confirms our finding that the left AG recombines schema components after consolidation.

**Table 5.** Transfer test: multi-voxel representations of schema components.

| Brain region | MNI | | | Z value | Cluster size |
|---|---|---|---|---|---|
| | x | y | z | | |
| **MVPA transfer test, rule-based associations** | | | | | |
| L superior occipital gyrus | -15 | -78 | 22 | 3.63 | 211 |
| L angular gyrus | -35 | -72 | 35 | 3.57 | |
| L superior occipital gyrus | -12 | -90 | 20 | 3.23 | |
| **MVPA transfer test, low-level visual features** | | | | | |
| L lingual gyrus | -12 | -82 | 8 | | 11692 |

Clusters that significantly discriminated schema component representations (rule-based associations, low-level visual features) during the transfer test. Bold font indicates the type of MVPA analysis. MNI coordinates represent the location of peak voxels. We report the first three local maxima (>8 mm apart) within each cluster (rule-based associations), and the local maximum for the low-level visual features analysis. Effects were tested for significance using cluster-inference with a cluster-defining threshold of $P < 0.005$ (rule-based associations) or $P < 0.001$ (low-level visual features) and a cluster-probability of $P < 0.05$ family-wise error (FWE) corrected for multiple comparisons (critical cluster sizes: rule-based associations, 172 voxels; low-level visual features, 60 voxels). L – left, R – right.

## Discussion

In this study, we investigated the retrieval dynamics of well-controlled, rule-based schemas and identified representations of their constituting components. These components consisted of rule-based associations and low-level visual features. Most importantly, both levels of information converged within the left AG after 24-hour-consolidation.

Memory networks are subject to reconfiguration as consolidation progresses. This process promotes the involvement of neocortical structures relevant for schema operations while downscaling MTL engagement (*Frankland and Bontempi, 2005*; *Takashima et al., 2006*; *Takehara-Nishiuchi and McNaughton, 2008*), possibly reflecting the abstraction and integration of information into pre-existing knowledge structures (*Lewis and Durrant, 2011*). To start out, we observed increased activation within MPFC, PCC, and higher-level sensory cortices during schema retrieval after 24 hours (*Figure 3C*). The MPFC is considered to play a pivotal role for schema-related mnemonic function (*Tse et al., 2011*; *Kroes and Fernandez, 2012*), which is supported by lesion studies in both rodents (*Richards et al., 2014*) and humans (*Ghosh et al., 2014*; *Warren et al., 2014*). Also the PCC (including precuneus and retrosplenial cortex), is regarded as central to memory processes (*Maguire et al., 1999*; for a review, see *Vann et al., 2009*). Together, the MPFC, PCC, MTL, and AG constitute a network of brain regions that act in concert during retrieval of (episodic) memories (*Rugg and Vilberg, 2013*; *Watrous et al., 2013*; *King et al., 2015*). Here, we observed that this network is also associated with the retrieval of rule-based schema memories (*Figure 4*). Considering the associative character of both, episodic and schema memory, common neural substrates seem plausible. Similar to previous schema studies with human subjects (*van Kesteren et al., 2010a*; *van Buuren et al., 2014*), we did not find a disengagement of hippocampal activation during retrieval of consolidated schema material. However, the hippocampus showed increased coupling with the retrieval network across days. Additionally, and in line with previous results (*Takashima et al., 2009*), we found a decrease in hippocampal-neocortical coupling after 24 hours (**Materials and methods, Complementary analysis: hippocampal connectivity during schema retrieval**).

The different components of a schema memory are assumed to be stored as distributed signatures (*Bartlett, 1932*; *Schacter et al., 1998*; *Wang and Morris, 2010*). At the same time, such a distributed memory system argues for the need to "bind" information in order to merge and recombine associative schema components upon retrieval. The novel feature of our experimental design allowed us to isolate the different schema components that consisted of rule-based associations and low-level visual features of the task material, while controlling for various, potentially confounding factors (such as complexity and attentional demands; see *Guerin et al., 2012*). We found

that rule-based associations were represented in the left AG and right ventrolateral prefrontal cortex – the latter potentially imposing top-down control on rule-based retrieval mechanisms (*Reverberi et al., 2012*). Low-level visual features of the task material were represented in occipital regions, AG, and fusiform gyrus. Crucially, both schema components converged within the left AG on day 2 (*Figure 5A*). Their multi-voxel representations, however, generalized only partly across days (*Figure 5B*), and the AG did not recombine schema components during retrieval on day 1 (**Materials and methods, Complementary analysis: AG involvement in schema retrieval on day 1**). That is, while low-level visual features showed shared representations between days and were detectable on day 1, representations of rule-based associations were not. Although the lower amount of retrieval trials on day 1 and the classification across two separate fMRI sessions might have affected the analyses, the coherence of our results suggests a change in the underlying representations, in particular for rule-based associations, that emerges after 24-hour-consolidation. Therefore, we conclude that the AG supports the integration of consolidated schema components during retrieval. This is corroborated by studies showing increased involvement of a parietal network in the processing of remote mnemonic content (for a review, see *Gilmore et al., 2015*).

Apart from theories that discuss the role of the AG in terms of mnemonic search and decision making (*Wagner et al., 2005*; *Gilmore et al., 2015*), the AG has been related to the "binding" of information. This is suggested by feature-integration theory (*Treisman and Gelade, 1980*), "cortical binding of relational activity" (CoBRA; a model presented by *Shimamura, 2011*), or accounts that identify the AG as heteromodal association cortex that recombines semantic information (*Binder et al., 2009*). A recent study by *Price and colleagues (2015)* demonstrated that the combination of semantic concepts is modulated by activation in the left AG (e.g., more activation for meaningful than non-meaningful combinations), and that subjects with lower cortical thickness in this region perform worse in this combinatorial task (*Price et al., 2015*). Further support for the "binding" notion comes from lesion studies. Typically, the impact of focal AG lesions is subtle. While patients with parietal lesions perform equally well as healthy controls in a recall task (*Simons et al., 2008*), a disruption of angular gyrus processing by transcranial magnetic stimulation reduces confidence (*Yazar et al., 2014*). Additionally, lesions in parietal cortex can cause so-called "illusory-conjunctions errors" where previously studied objects are identified, but mistakes are made when recombining information (*Friedman-Hill et al., 1995*; *Kesner, 2012*). This pattern of findings (intact retrieval, but impaired confidence and recombination of information) might be explained by limited damage to a distributed network that stores different memory components in respective brain structures (*Bartlett, 1932*; *Schacter et al., 1998*; *Wang and Morris, 2010*). In the present study, we showed that low-level visual features of the stimulus material were represented within the fusiform gyrus (*Figure 5A*). Additionally, the fusiform gyrus was functionally connected to the AG (*Figure 6*), as well as with the remaining retrieval network (*Figure 4*). This supports the assumption of strengthened cortico-cortical connections during schema retrieval (*Marr, 1970*; *Frankland and Bontempi, 2005*) that might act as a back-up in cases of AG disruption. If associative memory is truly dependent on the "binding" function of lateral parietal cortex, disruption should lead to an increase in memory conjunctions errors. Thus, memories should be retrieved, but 1) recombined in an incorrect manner, or 2) the combination of different memory features should not be possible at all. Future research could test this by experimentally inducing memory conjunction errors (*Reinitz et al., 1992*).

We were not able to identify schema representations within the remaining regions of the retrieval network (MTL, MPFC, and PCC). However, information might be represented at a finer spatial scale that cannot be detected by MVPA as done here. Also, our scan parameters were not optimized for the decoding of representations within the MTL (for example, see *Hassabis et al., 2009*), and the repeated retrieval of schema memories might have decreased our power to detect representations in the MPFC (*Woolgar et al., 2011*). This region, for example, was previously shown to hold remote, retrieval-related representations of specific autobiographical memories (*Bonnici et al., 2012*). Further, hippocampal cells that were active during the encoding of contextual fear memories were shown to be reactivated during retrieval (*Tanaka et al., 2014*). Silencing these cells rendered memory retrieval impossible. In either case, autobiographical and contextual fear memories certainly differ from more abstract schema memories. With the nature of these memory representations being so different, schema memories might simply not be represented within the MPFC, PCC, or MTL during retrieval.

Across different studies, definitions of the term "schema" so far ranged from simple (*Preston and Eichenbaum, 2013*) and more complex, experimentally-controlled associations (*Tse et al., 2007*, *2011*; *van Buuren et al., 2014*), to schemas that required the integration of new information into pre-existing real-world knowledge (*van Kesteren et al., 2014*). *Van Kesteren and colleagues (2014)*, for example, assumed with their design that prior knowledge guides congruency judgments of object-scene pairs, which in turn influences schema memory. However, this prior knowledge is difficult to control for as it is highly individual and thus may additionally involve self-referential, autobiographical memory processing. Here, we defined schemas as artificial sets of rules (*Kumaran et al., 2009*). While other studies may have greater ecological validity (*Maguire et al., 1999*; *van Buuren et al., 2014*; *van Kesteren et al., 2014*), we explicitly tailored this task to enable our analysis. This constitutes a crucial and novel feature of our design. By training and testing subjects on schema material across consecutive days, we achieved near-ceiling performance that allowed us to reliably train and test a classifier. To the best of our knowledge, we are the first to dissociate the multi-voxel representations of different schema components and to demonstrate their convergence during retrieval. Lastly, we show that new but related trials during the transfer test are solved by applying the schemas (*Figure 8*) and take this as evidence that our material provided a mental framework for subjects, allowing the rapid assimilation of new, related information (*Tse et al., 2007*). This is an important point in which schemas differ from so-called "task-sets" (*Sakai and Passingham, 2006*; *Bengtsson et al., 2009*; *Collins and Frank, 2013*). The creation of, or integration into a "categorical structure" is where the essence of schema lies.

Taking into account the range of schema definitions, it is currently unclear where the border between simple sets of rules and schemas should be drawn. To determine if our approach potentially constitutes a schema, we applied a set of criteria that was recently proposed by *Ghosh & Gilboa (2014)*. According to them, the necessary features for a schema memory are: (1) an associative network structure, (2) formation on the basis of multiple episodes, (3) the lack of unit detail, and (4) adaptability. Based on these criteria, our approach provides a very basic form of schematic memory: (1) our material has an associative structure, although simple; (2) schemas are not defined based on specific episodic information, material is learned fast but across multiple instances; (3) specific features are predictive while others are not; and (4) schemas could be expanded and adapted to new material (*Figure 8*).

To conclude, we manipulated the content of well-controlled, rule-based schema memories and were able to probe the functional dynamics during retrieval. We identified distributed representations of schema components that comprised rule-based associations and low-level visual features. These components converged within the left AG. Most importantly, this retrieval-related convergence was found only after 24-hour-consolidation and when transferring consolidated schemas to new but related task material. As such, the left AG might fulfill a role similar to the hippocampus during the retrieval of recent episodic memories (*Frankland and Bontempi, 2005*). In essence, we substantially expand current models of memory retrieval and provide neuroimaging evidence for a mechanistic framework in which the left AG acts as a convergence zone that may support the integration of distributed schema components.

## Materials and methods

### Subjects

Thirty-seven neurologically healthy, right-handed subjects (23 female, age range = 18–29 years, mean = 22) volunteered in this study. Eleven subjects were excluded from the study due to failure to learn the correct schemas after the first session. In particular, these excluded subjects made the following assumption: Encoding circle pair 3 (*Figure 1B*, left part, third from top) depicts two circles in horizontal arrangement; yellow on the left, blue in the center. When applying the spatial schema, this circle pair would yield the outcome "sun". During a retrieval trial (for example, circle pair 3 in *Figure 1B*, right part, that shows a yellow circle on the upper left and a blue circle at the lower left), the correct answer should again be "sun". Thus, spatial retrieval trials can be solved by acknowledging the horizontal position of one of the circles. Instead, eleven subjects solved such trials by mentally rotating the horizontal circle pair between encoding and retrieval with a 90° angle. Consequently, the inferred (correct) trial outcome was "sun" (since "the yellow circle was placed left

of the blue circle"), or (incorrectly) "rain" (since "the yellow circle was placed right of the blue circle"). This strategy resulted in a large amount of incorrect answers to spatial schema trials (day 1, % correct retrieval responses, mean ± s.e.m.: excluded subjects: 45.8 ± 1.7; included subjects: 89.6 ± 3.4), while the non-spatial schema was learned correctly (excluded subjects: 79.7 ± 3.1; included subjects: 93.4 ± 1.1). We excluded subjects based on their poor performance, which perfectly correlated with incorrect written schema explanations (**Materials and methods, Procedure**).

Additionally, two subjects aborted the experiment during the first session, and one was excluded due to technical problems (power breakdown). This left 23 subjects for analyses (16 female, age range = 18–29 years, mean = 22). All subjects had normal or corrected-to-normal vision and gave written informed consent. The study was conducted according to protocol approved by the institutional review board (CMO Region Arnhem-Nijmegen, The Netherlands).

## Material and task

Subjects learned to apply two sets of rules (i.e. schemas; spatial, non-spatial) in a deterministic weather prediction task in which colored circle pairs were associated with a fictive weather outcome ("sun", "rain"). Circle pairs could be solved with two different schemas regarding 1) the horizontal position of one circle (spatial schema; for example, "a circle on the left predicts sun"), or 2) the color of one circle (non-spatial schema; for example, "a blue circle predicts rain"). Thus, identical circle pairs could yield different weather outcomes when applying either spatial or non-spatial schemas (*Figure 1B*; see **Materials and methods, Procedure** for specific instructions to the subjects). Colored circles were matched for size and color intensity, and formed two different stimulus sets (yellow, blue, red; or green, orange, pink). While one set was used for schema encoding and retrieval across day 1 and 2, the other set was presented during the transfer test at the end of day 2 (*Figure 1A*). The order of stimulus sets was balanced across subjects. Stimulus material was created with Adobe Illustrator CS4 (Adobe, Inc.) and stimulus presentation was controlled using the Psychophysics Toolbox (*Brainard, 1997*).

Colored circles were presented in pairs at two possible orientations on the screen (left, right), and formed four distinct circle pairs during encoding and retrieval trials of the experiment (four circle pairs during encoding trials and four circle pairs during retrieval trials; *Figure 1B*). All circle pairs were presented during the experiment. During encoding trials, circles were presented in horizontal pairs. To make a clear distinction between both trial types, circles were presented in vertical pairs during retrieval. Thus, retrieval trials required the application of schematic knowledge to related information. To control for perceptual input, we created horizontal and vertical circle pairs that matched the spatial layout of encoding and retrieval trials but consisted of two-colored circles (perceptual baseline; *Figure 1—figure supplement 2C*). Subjects were instructed that these control trials would not follow any underlying schema and the response they needed to make was marked randomly.

## Procedure

The experiment consisted of two fMRI sessions on consecutive days (*Figure 1A*), specifically designed for MVPA approaches (*Coutanche and Thompson-Schill, 2012*). Sessions were approximately 24 hours apart (± 2 hours). Prior to the first scan session subjects were instructed to pay attention to spatial or non-spatial features ("The spatial rule concerns the position of a certain object, whereas the non-spatial rule concerns the color of a certain object.") but the exact stimulus-schema-outcome mappings were not provided. Further, they received a short training and familiarization with randomized feedback. This was followed by seven runs inside the MR scanner, each lasting approximately 9 min. Each run was structured in eight blocks of five trials each, whereby two blocks of encoding trials were always followed by two blocks of retrieval trials. Encoding and retrieval blocks contained trials of either spatial or non-spatial schema types, with one perceptual control trial randomly intermixed. All runs during the experiment consisted of equal amounts of encoding and retrieval trials, spatial and non-spatial trials, and trials with "sun"/"rain" outcomes. After completing day 1, subjects were asked to give a short written explanation of the two schemas (for example, "Please describe the spatial rule in your own words."). Additionally, subjects were shown the different circle pairs and were asked to indicate the outcomes when applying one or the other schema. Answers were scored as correct if they contained the correct association between

schema, color/position, and outcome. During day 2 subjects completed seven runs of retrieval blocks only. This yielded a total of 560 trials across both sessions (280 trials on day 1, of which 140 were encoding trials; 280 retrieval trials on day 2). The transfer test took place at the end of day 2. It comprised two runs inside the MR scanner that contained both encoding and retrieval trials (same structure as on day 1, see above; 80 trials across two runs, of which 40 were encoding trials and 40 were retrieval trials). The stimulus set was changed into circle pairs with different colors while keeping the same pair-wise arrangement (**Materials and methods, Material and task**).

Each block, irrespective of trial (encoding, retrieval) or schema type (spatial, non-spatial) shared the same timing parameters (*Figure 1—figure supplement 2*). At the beginning of a block, subjects were cued to use a specific schema type for solving all following trials. This was indicated by the word "spatial"/"non-spatial" printed in white font on a black computer screen (2 s). After a variable delay of 1.5–2 s, circle pairs were presented (3 s) and subjects had to think of the associated weather outcome. Then, after another short delay of 1–1.5 s, the response options (indicated by the letters "S" ("sun") or "R" ("rain")) were shown (2 s). To prevent fixed response-to-outcome mappings, response positions were randomly switched and subjects had to make a button press with their left or right index fingers. For encoding trials the correct answer was shown (2 s) and the next trial started after a short delay (1.5–2 s). No feedback was presented during retrieval trials, but subjects were asked to rate their confidence instead. Here, the options "not sure"/"sure" were presented on the screen (2 s). The confidence option "sure" should be chosen when being approximately 90% confident that the previous response was correct. Perceptual control trials followed the same timing as all other trials. Here, one response option was marked. After the successive presentation of five trials a black screen was shown (10–12 s) and a new block started.

## Behavioral data analyses

Performance and RTs of schema encoding data were tested with a run (1 to 7) × schema (spatial, non-spatial) repeated measures ANOVA; retrieval performance, RTs, and confidence were each analyzed with a day (day 1, day 2) × run (1 to7) × schema (spatial, non-spatial) ANOVA for repeated measures. Significant interaction effects were investigated with post-hoc ANOVAs and paired-sample $t$-tests. For the transfer test, behavioral data of schema encoding and retrieval were analyzed as above, but employing run (1, 2) × schema (spatial, non-spatial) ANOVAs for repeated measures. Greenhouse-Geisser correction was applied when appropriate and alpha was set to 0.05 throughout.

## Schema encoding

Learning quickly increased performance (main effect of run: $F(2.5,42.9) = 11.6$, $P < 0.0005$), which did not differ between schemas (no main effect of schema: $P = 0.209$; no run × schema interaction: $P = 0.441$; *Figure 9A*). Similarly, RTs decreased across runs (main effect of run: $F(6,102) = 5.4$, $P < 0.0005$) and did not differ between conditions (no main effect of schema: $P = 0.927$; no run × schema interaction: $P = 0.360$; see *Figure 9B*).

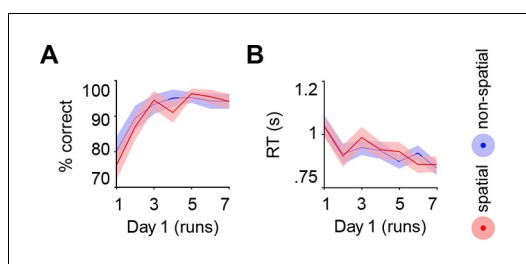

**Figure 9.** Behavioral performance during schema encoding. (A) Data represents the % of correct responses, and (B) the average reaction time (s). Shaded error bars denote ± s.e.m.

## Schema retrieval: reaction times

We found a significant day x run interaction ($F(2.9,49.5) = 4.8$, $P < 0.0005$; main effect of day: $F(1,17) = 4.6$, $P = 0.046$; main effect of run: $F(3.6,60.3) = 2.8$, $P = 0.038$; no main effect of schema: $P = 0.3$; no day × schema interaction: $P = 0.145$; no run × schema interaction: $P = 0.066$; no day × run × schema interaction: $P = 0.065$), followed up by separate run (1–7) × schema (spatial, non-spatial) ANOVAs for both days. Only for day 1 we found a significant main effect of run ($F(6,102) = 6.1$, $P < 0.0005$), and a run × schema interaction (interaction: $F(6,102) = 2.8$, $P = 0.013$; no main effect of schema: $P = 0.165$). Post-hoc paired-sample $t$-

tests revealed significantly shorter RTs for the non-spatial schema during run 1 ($t(21) = 2.5$, $P = 0.022$; *Figure 2B*, left). On day 2, we did not find any significant differences between runs or schema conditions (no main effect of run: $P = 0.718$; no main effect of schema: $P = 0.749$; no run × schema interaction: $P = 0.849$; *Figure 2B*, right).

## Schema retrieval: confidence

Retrieval was accompanied by a two-point confidence rating ("not sure"/"sure"). A three-way interaction between the factors day (day 1, day 2), run (1 to 7), and schema (spatial, non-spatial) ($F(3.8,65.3) = 4.2$, $P = 0.005$) was observed, suggesting that the difference in confidence ratings between days (main effect of day: $F(1,17) = 12.7$, $P = 0.002$) was caused by differences over runs or between schemas (main effect of run: $F(3,51.1) = 5.4$, $P = 0.003$; main effect of schema: $F(1,17) = 7.7$, $P = 0.013$; interaction day × run: $F(3,50.3) = 8.3$, $P < 0.0005$; interaction day × schema: $F(1,17) = 16.7$, $P = 0.001$; interaction run × schema: $F(6,102) = 3.7$, $P = 0.002$). To further test this, we employed a repeated measure ANOVA for each day with run and schema as factors. Only for day 1 we found an increase in retrieval confidence across runs (main effect of run: $F(2.7,45.4) = 7.5$, $P = 0.001$; main effect of schema: $F(1,170) = 12.2$, $P = 0.003$), and this increase differed between conditions (interaction run × schema: $F(3.4,58) = 4.8$, $P = 0.003$). Post-hoc paired-sample *t*-tests revealed lower confidence during retrieval of spatial, as compared to non-spatial, rule-based schema memories within the first run ($t(21) = -4.2$, $P < 0.0005$; *Figure 2C*, left). As can be seen, lower retrieval confidence for the spatial schema during the initial run of day 1 was also accompanied by lower retrieval performance and slower reaction times (*Figure 2A and B*, left). However, subjects quickly gained confidence. On day 2, retrieval confidence was at ceiling level and did not differ significantly between runs or schemas (no main effect of run: $P = 0.187$; no main effect of schema: $P = 0.397$; no run × schema interaction: $P = 0.549$; *Figure 2C*, right).

## Imaging parameters

Brain imaging data were acquired with a 3 Tesla MRI scanner (Trio Tim, Siemens, Erlangen, Germany) using a 32-channel head coil. For each run we obtained 256 $T_2$*-weighted BOLD images with the following parameters: gradient multi-echo EPI sequence (*Poser et al., 2006*), TR = 2100 ms, TEs = 7.6, 19.9, 32, 44 ms, flip angle = 80°, FOV = 200 × 200 mm, matrix = 80 × 80, 39 ascending axial slices, 10% slice gap, voxel size = 2.5 mm isotropic. Structural scans were acquired using a Magnetization-Prepared Rapid Gradient Echo (MP-RAGE) sequence with the following parameters: TR = 2300 ms, TE = 3.03 ms, flip angle = 8°, FOV = 256 × 256 mm, voxel size = 1 mm isotropic.

## fMRI data preprocessing

All imaging data were analyzed using SPM8 (http://www.fil.ion.ucl.ac.uk/spm/) in combination with Matlab (Matlab 2010b, The Mathworks, Inc., Natick, MA, USA). Due to technical problems with the gradient multi-echo EPI sequence, only echoes from echo-times 19.9, 32, 44 ms were used for analyses. Images from multiple echo-times were combined by first performing motion correction on the first echo (19.9 ms), estimating iterative rigid body realignment to minimize the residual sum of squares between the first echo of the first scan and all remaining scans. These estimated parameters were applied to all other echoes, thereby realigning all echoes to the first echo of the first scan. Then, the three echo images of each scan were combined into single images by calculating the weighted sum of the three echo times.

The first six volumes were discarded to allow for T1-equilibration. A total of seven runs in five subjects exceeded the limit of 2.5 mm movement and were excluded from further analysis. The combined EPI volumes from both fMRI sessions were slice-time corrected to the middle slice and realigned to the mean image. The structural image was co-registered to the mean image using mutual information optimization, and segmented into gray matter, white matter and cerebrospinal fluid. MVPA was performed in the native space of each subject. For univariate analysis, images were further spatially normalized to the Montreal Neurological Institute (MNI) EPI template using Diffeomorphic Anatomical Registration Through Exponentiated Lie Algebra (DARTEL; *Ashburner, 2007*), and smoothed with a 3D Gaussian kernel (8 mm full-width at half maximum, FWHM).

## Univariate activation analysis

The BOLD response for all correct trials of the different conditions was modeled as separate regressors time-locked to the onset of the presentation of the circle pairs (day 1: spatial encoding, non-spatial encoding, spatial retrieval, non-spatial retrieval; day 2: spatial retrieval, non-spatial retrieval). Additional regressors were included to model the perceptual baseline trials (day 2: perceptual baseline encoding, perceptual baseline retrieval; day 2: perceptual baseline retrieval), response periods (collapsed across response and feedback/confidence ratings), cues and incorrect trials (summarized as one regressor of no interest). All events were estimated as a boxcar function (circle pairs: 3 s, responses: 4 s, cues: 2 s) and convolved with a canonical hemodynamic response function (HRF). The efficiency of this design was verified prior to the start of the study, based on data from piloting. In addition, six realignment parameters and two regressors consisting of the mean signal of white matter and CSF were included in the design matrix. Next, a high-pass filter with a cutoff at 128 s was applied.

To address general effects of schema retrieval, we collapsed across spatial and non-spatial retrieval trials at a first level, created a general schema retrieval condition, and contrasted this against the perceptual baseline (schema retrieval > perceptual baseline). Schema consolidation was tested by entering these contrast images into a second level random-effects day (day 1, day 2) × run (1 to7) factorial design. Activation was tested for significance using cluster-inference with a cluster-defining threshold of $P < 0.001$ and a cluster-probability of $P < 0.05$ family-wise error (FWE) corrected for multiple comparisons.

## Connectivity analysis

We used Psychophysiological Interaction analyses (PPI; *Friston et al., 1997*) to probe functional coupling during schema retrieval. Two PPI analyses were performed per seed region (i.e. contrasts spatial schema retrieval > perceptual baseline and non-spatial schema retrieval > perceptual baseline). MPFC (x = -2, y = 35, z = -2) and PCC (x = 2, y = -45, z = 22) seeds (**Results, Schema retrieval networks: MPFC and PCC**) were defined as brain regions involved in the retrieval of consolidated schema memories (day 2 > day 1; consolidation contrast, **Results, Schema encoding**; *Figure 3C*), and coordinates of peak activations were chosen considering previous effects within these regions (*Kumaran et al., 2009*), as well as anatomical boundaries (*Nieuwenhuis and Takashima, 2011*). For these coordinates, a sphere with a radius of 8 mm was placed around the peak activations. The left AG seed (**Results, Schema convergence networks**) was delineated with a mask resulting from the convergence of schema components (1437 voxels; *Figure 5A*). The hippocampal seed region (**Materials and methods, Complementary analysis: hippocampal connectivity during schema retrieval**) was defined as bilateral hippocampus and was based on the Automatic Anatomical Labeling (AAL) atlas (http://fmri.wfubmc.edu/software/pickatlas). Next, time courses of each seed region were extracted. The interaction between time course and psychological factor (i.e., spatial schema retrieval > perceptual baseline × regional time course, and non-spatial schema retrieval > perceptual baseline × regional time course) was computed and activity positively related to this interaction was investigated. To test for group effects on day 2, individual contrast images were entered into a second level analysis and activity explained by the PPI regressor was tested with one-sample *t*-tests. To reveal the functional network involved in general retrieval processes, irrespective of the distinct rule-based schemas, we made a second-level conjunction (logical "and") of both ((spatial schema retrieval > perceptual baseline) ∩ (non-spatial schema retrieval > perceptual baseline); *Figure 4*, and **6**, shown in purple). Hippocampal connectivity was tested for changes over time and individual contrast images were thus submitted to a second level random-effects day (day 1, day 2) × run (1 to7) × schema (spatial, non-spatial) factorial design. All effects were tested for significance using cluster-inference with a cluster-defining threshold of $P < 0.001$ and a cluster-probability of $P < 0.05$ family-wise error (FWE) corrected for multiple comparisons.

## Multi-voxel pattern analysis

We started out by obtaining single-trial parameter estimates for later classification analyses. To this end, each schema retrieval trial was modeled as a separate regressor (*Mumford et al., 2012*) with remaining regressors appended identically to our first level estimation for univariate analysis (see above). Runs were modeled independently. This yielded 140 single-trial *t*-maps for day 1 and 280

single-trial *t*-maps for day 2 per subject. For the transfer test, all trials (encoding and retrieval) were included in the analysis due to the limited amount of data. This resulted in 64 single-trial *t*-maps per subject. To dissociate the distributed representations of schema components, we implemented MVPA using the library for support vector machines (LIBSVM, http://www.csie.ntu.edu.tw/~cjlin/libsvm/). For all MVPA analyses, a spherical searchlight (*Kriegeskorte et al., 2006*) was centered at each voxel in turn, considering all surrounding voxels within a radius of 8 mm. Only searchlights that included more than 30 gray matter voxels were examined. Features (i.e., voxels) were transformed into a pattern vector and a linear SVM classifier with a fixed regularization parameter $C = 1$ was trained to discriminate between schema components that consisted of (1) rule-based associations, and (2) low-level visual features of the task material (*Figure 1—figure supplement 1A*). The Matlab code for all searchlight analyses is openly available via https://github.com/isabellawagner/searchlight-svm.

First, we reasoned that subjects should show stable multi-voxel brain patterns of rule-based schema memories on day 2. Training and testing a classifier on this data would thus allow us to identify the neural signatures of both schema components (*Figure 1—figure supplement 1B*). Only correct and high-confidence data from day 2 was used for training and testing (spatial vs. non-spatial; number of trials per category, mean ± s.d.: 104 ± 9 vs. 105 ± 7). Discrimination performance was assessed using a 7-fold cross-validation regime during which the classifier was trained on data from six runs and tested on the remaining. This was repeated until every independent run was tested once. The average discrimination performance of each searchlight was assigned to its center voxel. Additionally, we trained a classifier to determine brain regions that solely distinguished between low-level visual features of the circle pairs rather than rule-based schema representations. Visual features necessary to predict a certain trial outcome consisted of the 1) position (spatial rule), and 2) color (non-spatial rule) of one of the circles. For retrieval trials, this information was not orthogonal in our experiment (a specific color appeared always on the left). By discriminating between the different circle pairs (circle pairs 1 and 2 vs. circle pairs 3 and 4; *Figure 1—figure supplement 1A*; number of trials per category, mean ± s.d.: 104 ± 8 vs. 105 ± 7) we were able to capture the visual features that differed between them. Classifier predictions were obtained as described above.

After completing the discrimination procedure for all possible searchlights within a volume, a 3D performance map was created. Individual performance maps were corrected for chance level by subtracting 50% (binary discrimination) from every voxel. Performance above chance implied the presence of discriminative information within this local voxel-pattern. Maps were normalized using DARTEL and smoothed with a 3 mm Gaussian kernel (FWHM). To test for statistical significance at a group-level, we submitted individual performance maps to a one-sample *t*-test in SPM8. Effects were tested for significance using cluster-inference with a cluster-defining threshold of $P < 0.001$ and a cluster-probability of $P < 0.05$ family-wise error (FWE) corrected for multiple comparisons.

As a next step, we investigated the generalization of multi-voxel representations across days and study phases. We repeated the training step, again using only correct and high-confidence data of day 2 (see above; discrimination between rule-based associations and between low-level visual features). However, we tested the classifier on neural data from day 1 (*Figure 1—figure supplement 1C*) and the transfer test (*Figure 1—figure supplement 1D*). Discrimination performance significantly above chance level would thus indicate shared multi-voxel representations of schema components between day 2 and day 1/the transfer test.

Throughout day 1, subjects completed 7 runs that each contained 16 retrieval trials (**Materials and methods, Procedure**). Since retrieval performance, RTs, and confidence across day 1 increased quickly and differed between runs (*Figure 2*), we predicted every run separately. Due to the small amount of data per run, we included all retrieval data (disregarding correctness or confidence). The resulting whole-brain maps were post-processed (see above) and submitted to a second level ANOVA with run (1 to 7) as within-subjects factor.

The transfer test consisted of two runs at the end of day 2 that each contained encoding and retrieval trials (8 trials each). We included thus all trials for testing the classifier (disregarding the trial type, correctness, retrieval confidence, or run). As above, individual discrimination maps were post-processed and submitted to one-sample *t*-tests. Again, unless stated otherwise, all effects were tested for significance using cluster-inference with a cluster-defining threshold of $P < 0.001$ and a cluster-probability of $P < 0.05$ family-wise error (FWE) corrected for multiple comparisons.

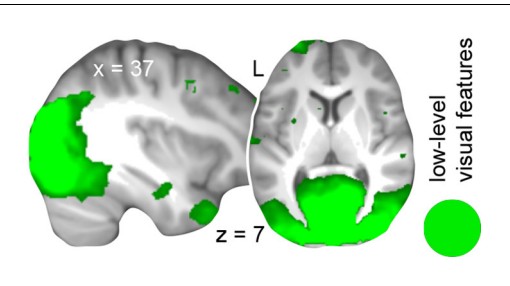

**Figure 10.** Multi-voxel representations of low-level visual features on day 1. Additional searchlight MVPA revealed distributed representations of low-level visual features, but not rule-based associations, during schema retrieval on day 1 (**Materials and methods, Complementary analysis: AG involvement in schema retrieval on day 1**). For display purposes, the map was resliced to a voxel dimension of 0.5 mm isotropic and is shown at P < 0.001, uncorrected. Significant clusters are noted in *Table 6*. L – left.

## Complementary analysis: AG involvement in schema retrieval on day 1

We performed additional MVPA analyses to test the AG involvement during schema retrieval on day 1. If schema component representations converged within the left AG during retrieval on day 1, this would point towards a general involvement of the AG, rather than a schema-specific involvement after 24-hour-consolidation. Therefore, we mimicked the previous MVPA of day 2 data. This contained the training and testing of two classifiers for each local searchlight pattern, using 7-fold cross-validation (fully described in **Materials and methods, Multi-voxel pattern analysis**; rule-based associations: spatial vs. non-spatial; low-level visual features: circle pairs 1 and 2 vs. circle pairs 3 and 4; trials per category, mean ± s.d.: 54 ± 5). Since day 1 contained only small amounts of retrieval trials (8 trials per condition, per run), we included all retrieval trials in the analysis (irrespective of correctness or retrieval confidence). As for the day 2 analysis, individual performance maps were entered into one-sample *t*-tests in SPM8. Unless stated otherwise, effects were tested for significance using cluster-inference with a cluster-defining threshold of P < 0.001 and a cluster-probability of P < 0.05 family-wise error (FWE) corrected for multiple comparisons.

We did not find any significant representations of rule-based associations within the AG or any other brain region on day 1. However, low-level visual features were represented as expected within occipital regions, extending into the AG, as well as within the right anterior temporal lobe (*Figure 10*; *Table 6*, upper part). Results appeared similar when we repeated this analysis in 14 subjects, selecting only correct retrieval trials with high confidence ratings (excluding nine subjects that

**Table 6.** Multi-voxel representations of low-level visual features on day 1.

| Brain region | MNI | | | Z value | Cluster size |
|---|---|---|---|---|---|
| | x | y | z | | |
| **MVPA day 1, all trials, N = 23** | | | | | |
| R middle occipital gyrus | 12 | -88 | 10 | | 16470 |
| R inferior temporal gyrus | 40 | 12 | -40 | 3.92 | 180 |
| **MVPA day 1, correct and high confidence trials, N = 14** | | | | | |
| R middle occipital gyrus | 12 | -88 | 10 | 6.24 | 12975 |
| L postcentral gyrus | -52 | -22 | 40 | 4.32 | 83 |
| L insular cortex | -35 | 10 | 2 | 4.27 | 75 |
| R inferior frontal gyrus | 50 | 5 | 0 | 3.88 | 91 |

Clusters that significantly discriminated the low-level visual features during retrieval on day 1. Bold font indicates the type of MVPA analysis (day 1, training and testing the classifier on day 1 using cross-validation; **Materials and methods, Complementary analysis: AG involvement in schema retrieval on day 1**). MNI coordinates represent the location of peak voxels. We report the local maximum of each cluster. Effects were tested for significance using cluster-inference with a cluster-defining threshold of P < 0.001 and a cluster-probability of P < 0.05 family-wise error (FWE) corrected for multiple comparisons (critical cluster sizes: upper part, 80 voxels; lower part, 65 voxels). L – left, R – right.

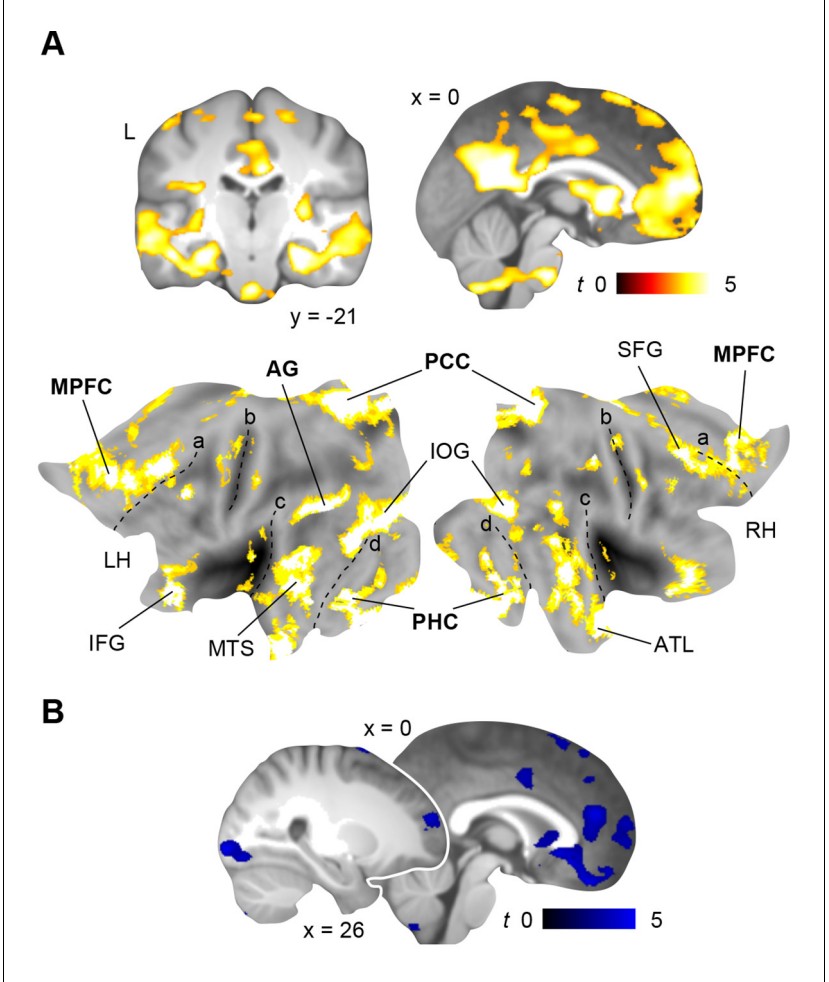

**Figure 11.** Hippocampal connectivity during schema retrieval. (**A**) Hippocampal connectivity during general schema retrieval (compared to the perceptual baseline) across both days (day 1 & day 2). Additionally, connectivity results are projected onto a surface-based flatmap. Relevant structures are labeled: AG, angular gyrus; ATL, anterior temporal lobe; IFG, inferior frontal gyrus; IOG, inferior occipital gyrus; MPFC, medial prefrontal cortex; MTS, medial temporal sulcus; PCC, posterior cingulate cortex; PHC, parahippocampal cortex; SFG, superior frontal gyrus. Regions of the retrieval network are highlighted in bold font. Dashed lines are inserted to aid orientation: a, border between medial and lateral prefrontal cortices; b, central sulcus; c, superior temporal gyrus; d, border between ventromedial and -lateral temporal cortices. LH – left hemisphere, RH – right hemisphere. (**B**) Decreased hippocampal-neocortical coupling during schema retrieval from day 1 to day 2 (day 1 > day 2). For display purposes, all maps were resliced to a voxel dimension of 0.5 mm isotropic and are shown at $P < 0.001$, uncorrected. Significant clusters are noted in *Table 7*. L – left.

showed one or more runs without correct and high confidence trials on day 1; trials per category, mean ± s.d.: rule-based associations, 48 ± 7 vs. 50 ± 7; low-level visual features, 48 ± 8 vs. 50 ± 6). Again, we did not find significant representations of rule-based associations, and low-level visual features were mostly represented in occipital regions (*Table 6*, lower part).

## Complementary analysis: hippocampal connectivity during schema retrieval

First, we assessed retrieval effects across both days and schema conditions. Results showed increased functional coupling between the hippocampus and an extensive set of regions, comprising surrounding MTL structures, the MPFC, PCC, and lateral occipital cortex (*Figure 11A*; *Table 7*, upper part). We did not find a difference in hippocampal coupling between the two schema

**Table 7.** Hippocampal connectivity during schema retrieval.

| Brain region | MNI | | | Z value | Cluster size |
|---|---|---|---|---|---|
| | x | y | z | | |
| Day 1 & day 2 | | | | | |
| R precuneus | 2 | -55 | 18 | 7.06 | 18139 |
| Cerebellum | -5 | -52 | -45 | 6.04 | 602 |
| R angular gyrus | 52 | -60 | 30 | 4.84 | 339 |
| R superior parietal gyrus | 28 | -38 | 58 | 4.62 | 139 |
| L insular cortex | -32 | -20 | 20 | 4.57 | 90 |
| R middle occipital gyrus | 32 | -80 | 40 | 3.94 | 111 |
| Day 1 > day 2 | | | | | |
| R cingulate gyrus | 2 | 22 | -5 | 5.12 | 162 |
| R middle occipital gyrus | 35 | -90 | 0 | 4.91 | 279 |
| L middle occipital gyrus | -38 | -88 | -2 | 4.43 | 295 |
| R superior frontal gyrus | 28 | 5 | 70 | 4.23 | 108 |
| R middle frontal gyrus | 22 | 58 | 18 | 4.01 | 144 |
| R cingulate gyrus | 5 | 40 | 15 | 4.01 | 87 |

Clusters that showed a significant increase in hippocampal connectivity during schema retrieval (**Materials and methods, Complementary analysis: hippocampal connectivity during schema retrieval**). Bold font indicates contrasts. Retrieval (collapsed across spatial and non-spatial schema conditions) was compared to the perceptual baseline. MNI coordinates represent the location of peak voxels. We report the local maximum of each cluster. Effects were tested for significance using cluster-inference with a cluster-defining threshold of $P < 0.001$ and a cluster-probability of $P < 0.05$ family-wise error (FWE) corrected for multiple comparisons (critical cluster size = 76 voxels). L – left, R – right.

conditions (no main effect of schema). Second, to investigate time effects in hippocampal connectivity, we chose a specific contrast between the days that allowed us to equate for differences in retrieval performance, confidence, and reaction times (day 1, runs 5–7 vs. day 2, run 1; **Results, Schema consolidation**). Results showed decreased coupling between the bilateral hippocampus and MPFC, as well as lateral occipital cortex during schema retrieval on day 2 as compared to day 1 (day 1 > day 2; *Figure 11B*; *Table 7*, lower part). No region showed increased functional coupling with the hippocampus during retrieval on day 2 as compared to day 1 (day 2 > day 1).

## Acknowledgements

The authors would like to thank the reviewers for their helpful comments, Alejandro Vincente-Grabovetsky for valuable advice on the analysis, Linda de Voogd for inspiring discussions, and Ruud Berkers for assistance in data acquisition. This project was supported by a grant from the European Research Council (ERC R0001075) to Richard G. Morris and Guillén Fernández.

# Additional information

## Funding

| Funder | Grant reference number | Author |
|---|---|---|
| European Research Council | R0001075 | Richard G Morris<br>Guillén Fernández |

The funders had no role in study design, data collection and interpretation, or the decision to submit the work for publication.

## Author contributions

ICW, Conception and design, Acquisition of data, Analysis and interpretation of data, Drafting or revising the article; MvanB, TPG, Analysis and interpretation of data, Drafting or revising the article; MCWK, MvanderL, RGM, Conception and design, Drafting or revising the article; GF, Conception and design, Analysis and interpretation of data, Drafting or revising the article

## Ethics

Human subjects: All subjects gave written informed consent prior to participation. The study was conducted according to protocol approved by the institutional review board (CMO Region Arnhem-Nijmegen, The Netherlands).

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
