## [Decision Letter]

Thank you for submitting your work entitled "Schematic memory components converge within angular gyrus during retrieval" for peer review at *eLife*. Your submission has been favorably evaluated by Timothy Behrens (Senior editor) and three reviewers, one of whom is a member of our Board of Reviewing Editors.

The reviewers have discussed the reviews with one another and the Reviewing editor has drafted this decision to help you prepare a revised submission.

Summary:

Wagner and colleagues present an fMRI study designed to reveal the neocortical networks involved in schema memories. They use a novel 'schema' paradigm that requires participants to utilize different task-rules to identical stimuli, focussing on fMRI activity on the day following initial learning. Across univariate, multivariate and functional connectivity analyses they identify a network of regions associated with schema processes. However, their data point to the angular gyrus as playing a privileged role – it "appears to recombine the different schema components into one memory representation".

Essential revisions:

1) There is a lack of discussion or detailed analysis of the MTL or hippocampus, even though studies on schemas have previously highlighted the role of the hippocampus in both spatial (Morris) and combined spatial-nonspatial (Mckenzie et al. Neuron 2014) schemas. Is the hippocampus (or MTL) involved in, or act as a hub either early or later over time? Simply showing hippocampal activation across types of schemas is not satisfactory (Figure 3).

2) The authors describe the task somewhat briefly at several points in the manuscript, a detailed description is lacking. For the yellow/blue/red group, was yellow always presented? If so, and blue was the color that related to "sun" does this mean red was always related to "rain"? A written description of the exact pairings at encoding and retrieval would help a lot. Also, at retrieval not all pairings are used it seems, so this again needs to be made clear (it may be mentioned in relation to the MVPA analyses).

3) The authors present a clear definition of schema and relate their task to this definition. However, I couldn't help thinking "isn't this just a task-set", or at least a consolidated task-set? This may well still be a "schema" according to their definition, but it seems very different from the spatial tasks used by Tse/Morris and the semantic congruency manipulations used by Van Kesteren/Fernandez. To what extent can the results be applied to "schema" in general as opposed to the specific type of task-set related schema specifically used here?

4) The MVPA analyses distinguished between the two tasks and visual features. They relate these effects to schemas, however they do not show the angular gyrus effect isn't present on day 1 (the only day 1 vs. 2 comparison is a univariate analysis). If the MVPA approach also distinguishes between the task and/or visual features on day 1, presumably this undermines their account in relation to consolidated schemas and suggests the angular gyrus is involved in retrieval in general (regardless of consolidation)? This would fit with the fMRI work (cited in the paper) showing lateral parietal effects during retrieval of episodic memory (importantly, usually retrieved on the same day as encoding).

5) By operationalizing schemas as conceptual rules (even more simplified than Kumaran et al. (2009), in which subjects had to learn spatial-fractal or fractal-fractal conjunctions without explicit instruction as to the rule; in the current task they only have to learn right/left and color mappings, and are provided with instructions about which type of rule to apply), how does this study improve the understanding of "schema" acquisition and retrieval, as opposed to goal-directed/rule-based learning? Specifically, the current task could be performed by directing attention to different stimulus dimensions (color or location) depending on the explicitly indicated rule, and it is unclear how these findings would apply to situations beyond this task and thus inform our understanding of schema more generally.

6) Interpretation of the MVPA results requires a more thorough definition of schema units. Ghosh and Gilboa (2014) cite Bartlett (1932) saying: "Schemas are general, higher-level constructs that encompass representations of the similarities or commonalities across events, rather than the specificity that make those events unique. This property was perhaps best articulated by Bartlett (1932), who said ‘the past acts as an organized mass rather than as a group of elements each of which retains its specific character’." If schemas are formed from multiple encoding episodes (and thus "frameworks" abstracted from specific experiences), then why is the specific low-level visual information a "unit" of the schema? The low-level visual MVPA analyses are likely revealing brain areas that discriminate between left/right visual fields (confounded with color: red/blue or orange/pink), so it should be clarified explicitly how this visual information would be a unit of these particular schema, why this visual content would be retrieved as part of the schema during the task (and not reflecting what is visually presented), and further, what it means for this particular visual information to be integrated/recombined with the abstracted rule during retrieval of the schema. However, aside from schemas, this finding could be interesting with respect to goal-directed/rule-based learning, particularly with regards to how abstracted rules might be integrated with perceptual information to make a decision.

7) Is it believed that subjects are undergoing "schema retrieval" on every trial of the retrieval task (especially given the number of trials they've completed by the beginning of day 2), or that subjects are actively maintaining the relevant well-learned rule throughout each block? Neural data from the transfer test might be particularly useful in looking at retrieval of the learned schemas, as the schemas would theoretically need to be retrieved in order to integrate the new information into the pre-existing knowledge. However, the nature of the transfer task seems like it would entail rule-updating only for the non-spatial task (learning new colors), rather than the spatial task (the left/right rule still applies); this may explain why transfer was faster for the spatial task.

8) While the MVPA analyses revealed regions that could discriminate between rules and visual stimuli relative to chance (i.e., 50% for binary classification), permutation tests would be a more rigorous test of these hypotheses, given the generally high levels of false positivity when performing voxel-wise-t-tests, and the often non-gaussianity of the accuracy probability distributions, etc. (e.g., Stelzer et al., 2013). Further, for each MVPA analysis (rule-based, visual) it would be informative to look into the SVM weight maps to see which voxels are more responsive for spatial vs. non-spatial trials, etc. Based on the interpretation that information about both spatial and non-spatial rules (and both visual "categories", e.g., left vs. right) is converging in angular gyrus, it would be helpful to show that these AG voxels are not selective for one type of information.

9) For the behavioral analyses, it appears that the ANOVAs were conducted treating run as a categorical variable. While not critical for interpretation of the results, it would be more appropriate to treat run as a continuous variable, and test for a quadratic effect of run; simple effect analyses could then be used, e.g., to test for an effect of schema at run 1 if there is a significant run x schema interaction. Further, is it warranted to treat the retrieval runs on day 2 as from 1-7? Couldn't these be interpreted as runs 8-14? Finally, for transfer test analysis (it was unclear from the methods when the transfer test occurred, presumably on day 2; was this scanned?), if trying to show that subjects were applying knowledge learned on day 1 (and thus transfer from learning on day 1), effectively using the "schemas" to learn more quickly, the analyses should be comparing performance on the transfer task to the relative initial learning runs from day 1, rather than within the two runs of the transfer test.

Minor comments [abridged]:

1) The authors say: "response positions were balanced within each run". What does this mean? That each trial gave a different response-to-outcome mapping (i.e., changed S/R responses with regard to left and right hand across trials)?

2) The description of transfer test is confusing: "The stimulus set was changed into circles with different colors while keeping the same pair-wise arrangement". What exactly does this mean? You mention the same colors after this (e.g., yellow/blue/red). Does this mean you change what color is associated with the location/color condition respectively so participants have to apply the same task to a new color/location combination?

3) In the subsection “Univariate activation analysis”, please clarify what you mean by "i.e. conjunction contrast". You are contrasting the average of two conditions vs. a further condition; you are not conducting a proper conjunction analysis (as in your MVPA analyses). To avoid confusion, we suggest you not call this a "conjunction contrast".

4) Why does the design require that both tasks rely on color, but only one on location (i.e., "sun when the yellow circle is on the left")? Although the other circle always coincides in location with the yellow circle, the task instructions are clear to pointing to a color. As such, it seems whereas the "color" task is purely related to "color" the "location" task relates to the location of a specific color. Was there a reason for not making the instructions purely location-based?

5) Why did you not orthogonalize the color/location pairs at retrieval (as you did at test), so you could then classify the "location" and "color" visual features separately? This seems like a stronger test of your AG effect – showing this region codes both task-related aspects.

6) Why is it "unlikely that convergence of schema components emerges, for example, from spatial blurring between two neighboring, functionally distinct regions"?

7) Retrieval performance was better for non-spatial than spatial on Day 1 – is this because of the rotation of items? The "rule" for the color (non-spatial) condition was the same as encoding with the modified stimuli.

8) Is there evidence for consolidation over 24 hours? Would the same network of regions appear just as a function of run within a day, or does this only emerge after sleep? Similarly, if consolidation is "a prerequisite for mental schemas," then how were subjects able to perform so well on the retrieval task (arguably requiring some transfer from the encoding task) on day 1?

9) For the AG PPI analyses, how was the seed determined? Only the MPFC and PCC seeds are defined in the Materials and methods, Connectivity analyses. It looks like it’s just the local maxima pulled from the rule-based schema MVPA analyses, but is that region overlapping with the visual-features voxels? Also, it appears that there are some other similar sized regions of overlap between both MVPA analyses in the right hemisphere, more medial; would it be possible to get a table of all conjunctions?

[Editors' note: further revisions were requested prior to acceptance, as described below.]

Thank you for resubmitting your work entitled "Schematic memory components converge within angular gyrus during retrieval" for further consideration at *eLife*. Your revised article has been favorably evaluated by Timothy Behrens (Senior editor) and a Reviewing editor. The manuscript has been improved but there are some remaining issues that need to be addressed before acceptance, as outlined below:

The authors have responded well to the first round of reviews and have clarified/improved the manuscript. However, there is one issue that was not adequately addressed. This concerned showing the AG was not involved on Day 1. To answer this, they trained a classifier on Day 2, and applied this to the data on Day 1. Whilst interesting, this doesn't address my core concern. They need to train/test on Day 1 (i.e., exactly the same analysis they do for the Day 2 data). If this reveals the AG, this would clearly undermine their conclusion in relation to "consolidated" schema. The classification from Day 2 retrieval to the transfer task on Day 2 is interesting, and at least shows some degree of generalization. However, there are many reasons why one might expect to see this effect, but not a Day 2 to Day 1 effect – not least because the latter is comparing across two completely separate scanning sessions.

---

## [Author Response]

Essential revisions:

*1) There is a lack of discussion or detailed analysis of the MTL or hippocampus, even though studies on schemas have previously highlighted the role of the hippocampus in both spatial (Morris) and combined spatial-nonspatial (Mckenzie et al. Neuron 2014) schemas. Is the hippocampus (or MTL) involved in, or act as a hub either early or later over time? Simply showing hippocampal activation across types of schemas is not satisfactory (Figure 3).*

In addition to our initial results that did not show changes in hippocampal activation over 24 hours (updated **Results, Schema consolidation**), we performed new analyses to clarify the role of the hippocampus in schema retrieval either early or later over time. We considered two aspects, activity and connectivity, because hippocampal contributions and the changes thereof can be reflected in differences in net activity, but also connectivity with neocortical structures relevant for memory retrieval (Takashima et al., 2009; van Kesteren et al., 2010). In line with Takashima and colleagues (2009), we revealed enhanced hippocampal connectivity with specific neocortical regions during retrieval of schema memories that decreased with time.

**A) Changes in hippocampal (and MTL) activity during schema retrieval and its changes over time:** Across both days, we found differences in parahippocampal but not hippocampal activation during schema retrieval (updated **Results, Schema consolidation**; updated Figure 3, and Figure 12). Comparing activity between the two days did not reveal any significant hippocampal (or MTL) activation differences (updated **Results, Schema consolidation**; updated Figure 3, and Figure 12).

Author response image 1.Hippocampal activation during schema retrieval (z = -17).(**A**) Increased BOLD responses during schema memory retrieval across both days (schema retrieval > perceptual baseline), (**B**) during rule-based schema retrieval on day 1 (day 1 > day 2), and (**C**) after an initial consolidation of 24 hours (day 2 > day 1). Contrasts include runs 5 to 7 from day 1, and run 1 from day 2. For display purposes, results were resliced to a voxel dimension of 0.5 mm isotropic and are shown at *P* < 0.001, uncorrected. Significant clusters are noted in Table 6 (updated manuscript). Results are superimposed onto the average structural scan derived from all subjects. The hippocampus is schematically outlined in black (based on the anatomical definition of the AAL atlas). L – left.**DOI:**
http://dx.doi.org/10.7554/eLife.09668.023

**B) Changes in hippocampal connectivity during schema retrieval over time:** Next, we tested for changes in hippocampal connectivity during schema retrieval using a Psychophysiological Interaction analysis (PPI). We used the anatomical masks of left and right hippocampus, provided by the Anatomical Automatic Labeling atlas (AAL; http://fmri.wfubmc.edu/software/pickatlas), and created a combined, bilateral hippocampus seed. Analysis steps were identical to our previous PPI analysis (**Materials and methods, Connectivity analysis**). In brief, we computed the interaction between the time course of the seed with the psychological factor (i.e., spatial schema retrieval > perceptual control × regional time course, and non-spatial schema retrieval > perceptual control × regional time course). Individual contrast images were entered into a second level random-effects day (day 1, day 2) × run (1 to 7) × schema (spatial, non-spatial) factorial design. Activation was tested for significance using cluster-inference with a cluster-defining threshold of *P* < 0.001 and a cluster-probability of *P* < 0.05 family-wise error (FWE) corrected for multiple comparisons (critical cluster size = 76 voxels).

First, we assessed retrieval effects across both days and schema conditions. Results showed increased functional coupling between bilateral hippocampus and an extensive set of regions, comprising surrounding MTL structures, the MPFC, PCC, and lateral occipital cortex (new Figure 10). There was no difference in hippocampal coupling between the two schema conditions (no main effect of schema). Second, to investigate time effects in hippocampal connectivity, we chose a specific contrast between the days that allowed us to equate for differences in retrieval performance, confidence, and reaction times (same as the consolidation contrast we reported in our original manuscript; day 1, runs 5-7 vs. day 2, run 1; **Results, Schema consolidation**). Results showed decreased coupling between the bilateral hippocampus and MPFC, as well as lateral occipital cortex during schema retrieval on day 2 as compared to day 1 (new Figure 10). No region showed increased hippocampal coupling on day 2 as compared to day 1.

**Summary and action taken:** We analyzed hippocampal activation and connectivity during schema retrieval. We found increased parahippocampal but not hippocampal activation across days; and further no hippocampal (or MTL) activation differences between days. Thus, in line with results from recentschema studies with human subjects we did not find a disengagement of hippocampal activation when retrieving "consolidated" schemas (van Kesteren et al., 2010; van Buuren et al., 2014). However, in line with previous results (Takashima et al., 2009), connectivity analysis revealed increased hippocampal-neocortical coupling during retrieval across both days, as well as decreased hippocampal-neocortical coupling on day 2 as compared to day 1.

We updated and moved the section Schema consolidation into the main Results section (**Results, Schema consolidation**), together with corresponding Figure 3 and Table 6. We further discuss the role of the hippocampus in our revised manuscript more thoroughly (**Discussion**) and included the additional analysis in the section **Materials and methods, Complementary analysis: hippocampal connectivity during schema retrieval**. Please see below for the edited text passages:

**Results, Schema consolidation:***“Across both days, retrieval was associated with increased BOLD responses in bilateral lingual gyrus, superior occipital gyrus, cuneus, left supplemental motor area, and right parahippocampal cortex.” (The remaining text is identical to our original manuscript.)*

**Discussion:***“Memory networks are subject to reconfiguration as consolidation progresses. This process promotes the involvement of neocortical structures relevant for schema operations while downscaling MTL engagement (Frankland and Bontempi, 2005; Takashima et al., 2006; Takehara-Nishiuchi and McNaughton, 2008), possibly reflecting the abstraction and integration of information into pre-existing knowledge structures (Lewis and Durrant, 2011). […] Similar to previous schema studies with human subjects (van Kesteren et al., 2010; van Buuren et al., 2014), we did not find a disengagement of hippocampal activation during retrieval of consolidated schema material. However, the hippocampus showed increased coupling with the retrieval network across days. Additionally, and in line with previous results (Takashima et al., 2009), we found a decrease in hippocampal-neocortical coupling after 24 hours (Figure 10; Table 6; Materials and methods, Complementary analysis: hippocampal connectivity during schema retrieval).”*

**Materials and methods, Complementary analysis: hippocampal connectivity during schema retrieval:***“First, we assessed retrieval effects across both days and schema conditions. […] No region showed increased functional coupling with the hippocampus during retrieval on day 2 as compared to day 1 (day 2 > day 1).”*

Finally, we also updated the PPI **Materials and methods**section:

**ID fig11 already definedMaterials and methods, Connectivity analysis:***“The hippocampal seed region (Materials and methods, Complementary analysis: hippocampal connectivity during schema retrieval) was defined as bilateral hippocampus and was based on the Automatic Anatomical Labeling (AAL) atlas (http://fmri.wfubmc.edu/software/pickatlas). (…) Hippocampal connectivity was tested for changes over time and individual contrast images were thus submitted to a second level random-effects day (day 1, day 2) × run (1 to7) × schema (spatial, non-spatial) factorial design.”*

*2) The authors describe the task somewhat briefly at several points in the manuscript, a detailed description is lacking. For the yellow/blue/red group, was yellow always presented? If so, and blue was the color that related to "sun" does this mean red was always related to "rain"? A written description of the exact pairings at encoding and retrieval would help a lot. Also, at retrieval not all pairings are used it seems, so this again needs to be made clear (it may be mentioned in relation to the MVPA analyses).*

We apologize for not explaining the task with enough detail. Indeed, subjects were always presented with the yellow circle if they were shown the yellow/blue/red stimulus set during encoding and retrieval across days. Stimulus material formed eight possible circle pairs (four encoding and four retrieval circle pairs, respectively). All possible circle pairs were used during encoding and retrieval. During the non-spatial schema condition, the color blue was related to “rain”, whereas red was always related to “sun” (for the yellow/blue/red stimulus set).

We now provide a more detailed description of the task in the section **Materials and methods, Material and task** to which we refer to at several points in the manuscript (for example, **Introduction**; see edited text below). Furthermore, we updated Figure 1 (including Figure 1—figure supplement 1, and Figure 1—figure supplement 2) which provides a complete picture of the experiment and task. This also includes a depiction of the exact pairings in the main Figure 1.

**Materials and methods, Material and task:***“Subjects learned to apply two sets of rules (i.e. schemas; spatial, non-spatial) in a deterministic weather prediction task in which colored circle pairs were associated with a fictive weather outcome (“sun”,”rain”). […] The order of stimulus sets was balanced across subjects. […] Colored circles were presented in pairs at two possible orientations on the screen (left, right), and formed four distinct circle pairs during encoding and retrieval trials of the experiment (four circle pairs during encoding trials and four circle pairs during retrieval trials; Figure 1). All circle pairs were presented during the experiment.”*

**Introduction:***“These schemas were incorporated into a modified, deterministic weather prediction task (Knowlton et al., 1994; Kumaran et al., 2009) in which subjects had learned that colored circle pairs predicted specific but fictive weather outcomes (“sun”, “rain”), depending on the location (spatial schema) or color (non-spatial schema) of one of the circles (Figure 1; for a detailed description please see Materials and methods, Material and task).”*

*3) The authors present a clear definition of schema and relate their task to this definition. However, I couldn't help thinking "isn't this just a task-set", or at least a consolidated task-set? This may well still be a "schema" according to their definition, but it seems very different from the spatial tasks used by Tse/Morris and the semantic congruency manipulations used by Van Kesteren/Fernandez. To what extent can the results be applied to "schema" in general as opposed to the specific type of task-set related schema specifically used here?*

Schemas have been broadly defined as relational knowledge structures that are applicable to a wide range of instances and which help to integrate new but related information (Bartlett, 1932; van Kesteren et al., 2012; Ghosh and Gilboa, 2014). However, this broad definition lead to different operationalizations, spanning widely from simple, rule-like associations (Preston and Eichenbaum, 2013) and more complex, visuo-spatial layouts (Tse et al., 2007, 2011; van Buuren et al., 2014), to semantic knowledge acquired throughout life(van Kesteren et al., 2014). Considering this spectrum of complexity, it remains an empirical question whether there is a clear border between simple sets of rules, or "task-sets" (Sakai and Passingham, 2006; Bengtsson et al., 2009; Collins and Frank, 2013), and schemas and, if so, where this border should be drawn (Kroes and Fernandez, 2012).

Schemas are indeed different from "task-sets" (Sakai and Passingham, 2006; Bengtsson et al., 2009; Collins and Frank, 2013), as they provide knowledge structures that help new and related information to be integrated more rapidly (Tse et al., 2007; McKenzie et al., 2014). Therefore, our schema material should allow the transfer to novel information. We tested this assumption using a transfer test for the non-spatial schema at the end of day 2 (see also our response to Point 9). The results showed that subjects were able to transfer the learned schema to novel trials successfully and did so even faster than during the initial acquisition on day 1 (Figure 13). Thus, we take this as indirect evidence that material acquired on day 1 provided a framework for the more rapid acquisition of novel information during the transfer test (these new results are incorporated in the section **Results, Transfer test: initial schema acqiuisition vs. new learning**). The creation of, or integration into such a conceptual framework is where the essence of schema lies.

Author response image 2.Non-spatial schema performance, RTs and retrieval confidence compared to day 1, run 1.(**A**) Schema Encoding: left, % of correct responses; right, average reaction time (**s**). (**B**) Schema retrieval: left, % of correct responses; middle, average reaction time (**s**); right, % of high-confident ratings (i.e. “sure”-responses). Error bars denote ± standard error of the mean (s.e.m.). ** marks significance at *P* < 0.001.**DOI:**
http://dx.doi.org/10.7554/eLife.09668.024

As also described in our original **Discussion**, we determined if our approach potentially constitutes a schema and applied a set of criteria that was recently proposed by Ghosh and Gilboa (2014). According to them, the necessary features for a schema memory are: (1) an associative network structure, (2) formation on the basis of multiple episodes, (3) the lack of unit detail, and (4) adaptability. Based on these criteria, our approach provides a very basic form of schematic memory: (1) our material has an associative structure, although simple; (2) schemas are not defined based on specific episodic information, material is learned fast but across multiple instances; (3) specific features are predictive while others are not; (4) schemas could be expanded and adapted to new material.

Naturally, more ecologically relevant approaches (Maguire et al., 1999; Tse et al., 2007, 2011; van Kesteren et al., 2010, 2014; McKenzie et al., 2014; van Buuren et al., 2014) may be closer to the intuitive notion of a "schema". However, such approaches do not allow the controlled dissociation of different schema components (here, rule-based associations and low-level visual features), as was possible with our design and MVPA analyses. A crucial and novel feature of this study is that we explicitly chose a simple and experimentally-controlled approach. By training and testing subjects on schema material across consecutive days, we achieved near-ceiling performance that enabled us to reliably train and test a classifier and to dissociate the multi-voxel patterns of both schema components (rule-based associations, low-level visual features).

**Summary and action taken:** We updated our Introduction and included a brief comparison between schemas and "task-sets" in the **Discussion** of our revised manuscript. Please find the edited text passages below.

**Introduction:***“So far, attempts to operationalize schemas spanned an entire spectrum […] and if so, where this border should be drawn (Kroes and Fernandez, 2012).”*

**Discussion:***“Lastly, we show that new but related trials during the transfer test are solved by applying the schemas (Figure 8) […]. The creation of, or integration into a ‘categorical structure’ is where the essence of schema lies.”*

*4) The MVPA analyses distinguished between the two tasks and visual features. They relate these effects to schemas, however they do not show the angular gyrus effect isn't present on day 1 (the only day 1 vs. 2 comparison is a univariate analysis). If the MVPA approach also distinguishes between the task and/or visual features on day 1, presumably this undermines their account in relation to consolidated schemas and suggests the angular gyrus is involved in retrieval in general (regardless of consolidation)? This would fit with the fMRI work (cited in the paper) showing lateral parietal effects during retrieval of episodic memory (importantly, usually retrieved on the same day as encoding).*

We performed additional analyses to investigate AG involvement during schema retrieval over time. We reasoned that if the neural signatures of converging schema components are created by consolidation processes, we should not be able to identify representations of rule-based associations on day 1, using a classifier that was trained on day 2 (summarized in Figure 1—figure supplement 1). Thus, we performed novel MVPA analyses to investigate how well multi-voxel patterns of schema components generalized across days.

Specifically, single-trials of day 1 were modeled as separate regressors, with remaining regressors appended identically to our first level estimation for univariate analysis. Runs were modeled independently. As in our original analysis (**Materials and methods, Multi-voxel pattern analysis**), a spherical searchlight (8 mm radius) was centered at every voxel in turn. First, for each of these local voxel patterns, we trained a classifier for rule-based associations (spatial vs. non-spatial) on correct and high-confident retrieval trials of day 2 and applied this classifier to all retrieval trials of day 1. Second, we repeated this procedure and trained a classifier for the low-level visual features of the task material (circle pairs 1 and 2 vs. circle pairs 3 and 4). This resulted in one whole-brain performance map per classifier, run, and subject. Images were entered into two repeated measures ANOVAs (one for each classifier) with run (1 to 7) as a within-subjects factor. Effects were tested for significance using cluster-inference with a cluster-defining threshold of *P* < 0.001 and a cluster-probability of *P* < 0.05 family-wise error (FWE) corrected for multiple comparisons (for a detailed description of analysis and statistical correction of performance maps please see **Materials and methods, Multi-voxel pattern analysis**).

We tested if any of the runs showed discrimination performance for rule-based associations significantly above chance level. We did not find significant discrimination performance across day 1 (Figure 14), indicating that the multi-voxel patterns of the two schema conditions were not shared between days (critical cluster size = 79 voxels). This null result remained also at lower cluster-defining thresholds (*P* < 0.005, critical cluster size = 178).

We defined the low-level visual features of the task material as a schema component since the visual features are connected to higher-order information (see also our reply to Point 6). However, circle patterns were visually presented on the screen. Thus, we reasoned that discrimination of low-level visual features should be possible across days and repeated the above analysis using a classifier that discriminated between the low-level visual features of the task material (**Materials and methods, Multi-voxel pattern analysis**). This analysis also served as a control to clarify that the generalization of multi-voxel patterns is not decreased due to methodological constraints such as differences in realignment of fMRI data from the different days. As expected, results showed significant discrimination performance in occipital cortex (Figure 14; critical cluster size = 70 voxels).

Author response image 3.Generalization of schema component representations from day 2 to day 1.(**A**) Multi-voxel patterns of rule-based associations did not generalize across days. (**B**) Multi-voxel patterns of low-level visual features were shared across days. For display purposes, all maps were resliced to a voxel dimension of 0.5 mm isotropic and are shown at *P* < 0.001, uncorrected. Significant clusters are noted in Table 3 (updated manuscript). L – left. This figure is incorporated in updated Figure 5.**DOI:**
http://dx.doi.org/10.7554/eLife.09668.025

**Summary and action taken:** In summary, we did not find significant generalization of multi-voxel patterns for rule-based associations across days. This indicates a difference in neuronal representations and we conclude that the AG seems to support the convergence of schema components only after a 24 hour-delay.

We acknowledge that this finding does not preclude AG involvement during retrieval on day 1. However, as also pointed out in our revised **Discussion** (see edited text below), the neural signatures of underlying processes might be different, leading to a null result in the generalization of rule-based association patterns between days. An alternative explanation is that the AG is only supports retrieval-related schema convergence after consolidation; which is in line with our results. This is also supported by studies showing increased involvement of a parietal network (including the AG) in the processing of remote mnemonic content (Gilmore et al., 2015).

We thank the reviewers for this excellent suggestion and are convinced that the novel results of our additional analyses significantly strengthen our point. We included this analysis in the **Results, Multi-voxel representations of schema components** (see text below), along with updated Figure 5 and Table 3 (updated manuscript).

**Results, Multi-voxel representations of schema components:**
*“Next, we reasoned that if schema components converged in the AG only after consolidation, the multi-voxel representations of rule-based associations should not generalize from day 2 to day 1. […] In summary, the left AG converged schema components only after a 24 hour-delay.”*

We further critically discuss the new results:

**Discussion:***“Crucially, both schema components converged within the left AG on day 2 (Figure 5); but not prior to the 24-hour-delay (Figure 5). […] This is supported by studies showing increased involvement of a parietal network in the processing of remote mnemonic content (for a review, see Gilmore et al., 2015).”*

Finally, we updated our **Materials and methods** section:

**Materials and methods, Multi-voxel pattern analysis:**
*“As a next step, we investigated the generalization of multi-voxel representations to day 1 and the transfer test. […] The resulting whole-brain maps were post-processed (see above) and submitted to a second level ANOVA with run (1 to 7) as within-subjects factor.”*

*5) By operationalizing schemas as conceptual rules (even more simplified than Kumaran et al. (2009), in which subjects had to learn spatial-fractal or fractal-fractal conjunctions without explicit instruction as to the rule; in the current task they only have to learn right/left and color mappings, and are provided with instructions about which type of rule to apply), how does this study improve the understanding of "schema" acquisition and retrieval, as opposed to goal-directed/rule-based learning? Specifically, the current task could be performed by directing attention to different stimulus dimensions (color or location) depending on the explicitly indicated rule, and it is unclear how these findings would apply to situations beyond this task and thus inform our understanding of schema more generally.*

Our response here is largely overlapping with our response to Point 3, because they are touching on a similar conceptual topic. Our task design represents indeed a simplified version of Kumaran and colleagues(2009). As already addressed above, more ecologically relevant approaches (Maguire et al., 1999; Tse et al., 2007, 2011; van Kesteren et al., 2010, 2014; McKenzie et al., 2014; van Buuren et al., 2014) may be closer to the intuitive notion of "schema", but these studies did not allow the dissociation of schema components. Furthermore, a number of previous studies, for example Van Kesteren and colleagues (2014), investigated how prior real-world knowledge guided congruency judgments and thus schema memory. Such prior knowledge is highly individual and may involve self-referential, autobiographical memory processing. We chose a simple task design to explicitly control for these effects.

By training and testing subjects on schema material across consecutive days, we achieved near-ceiling performance during day 2 (**Results, Behavioral performance**). Explicit instructions which schema to apply further supported our MVPA approach (multivariate discrimination methods such as Support Vector Machines benefit from a large number of trials, each preferably offering a stable representation of the data). These important design features enabled us to reliably train and test a classifier and to dissociate the multi-voxel patterns of both schema components (rule-based associations, low-level visual features). To the best of our knowledge, this is the first time the representational patterns of schema components were identified. Moreover, we showed that the region carrying and presumably "binding" these representations, namely the AG, is embedded in a network that has previously been assigned to memory retrieval, allowing for the first time the conclusion that schematic information is retrieved by the same network.

Importantly, we argue that our schema material differs from goal-directed learning or, for example, rule-based "task-sets" (Sakai and Passingham, 2006; Bengtsson et al., 2009; Kroes and Fernandez, 2012; Collins and Frank, 2013). We show that new but related trials are solved even faster during the transfer test as compared to initial schema acquisition on day 1 (new **Results, Transfer test new learning vs. initial schema acquisition**; see also Figure 13; and our response to Point 9). We take this as evidence that our schema material provided a mental framework for subjects that allowed the rapid assimilation of new and related information (Tse et al., 2007) – as opposed to goal-directed learning.

Indeed, the task could be performed by directing attention to spatial or non-spatial stimulus dimensions that were explicitly instructed. Nevertheless, these associations go beyond simple stimulus-response learning. Both schema conditions were based on identical visual input. Thus, automatic, attention-based prediction of the specific trial outcome could not have occurred. This could only be the case if visual input for the two conditions or the different schema outcomes was different.

**Action taken:** We understand the reviewers’ concern that our task is a simplification of "schema". However, simplifying schema material was a necessary step to understand the foundational mechanisms of schema retrieval. Regardless of this simplification, our schema material fulfills the necessary criteria for a "schema" (Ghosh and Gilboa, 2014), as also addressed in our original **Discussion**.

We have included these points in the updated **Discussion** of our revised manuscript (see edited text passages below) and hope that we have made it more apparent to the reviewers how exactly our findings could inform the understanding of schema more generally.

**Discussion:**
*“While other studies may have greater ecological validity (Maguire et al., 1999; van Buuren et al., 2014; van Kesteren et al., 2014), we explicitly tailored this task to enable our analysis. […] The creation of, or integration into a ‘categorical structure’ is where the essence of schema lies.”*

6) Interpretation of the MVPA results requires a more thorough definition of schema units. Ghosh and Gilboa (2014) cite Bartlett (1932) saying: "Schemas are general, higher-level constructs that encompass representations of the similarities or commonalities across events, rather than the specificity that make those events unique. This property was perhaps best articulated by Bartlett (1932), who said ‘the past acts as an organized mass rather than as a group of elements each of which retains its specific character’." If schemas are formed from multiple encoding episodes (and thus "frameworks" abstracted from specific experiences), then why is the specific low-level visual information a "unit" of the schema? The low-level visual MVPA analyses are likely revealing brain areas that discriminate between left/right visual fields (confounded with color: red/blue or orange/pink), so it should be clarified explicitly how this visual information would be a unit of these particular schema, why this visual content would be retrieved as part of the schema during the task (and not reflecting what is visually presented), and further, what it means for this particular visual information to be integrated/recombined with the abstracted rule during retrieval of the schema. However, aside from schemas, this finding could be interesting with respect to goal-directed/rule-based learning, particularly with regards to how abstracted rules might be integrated with perceptual information to make a decision.

Schemas consist of interrelated "units", or "features" (van Kesteren et al., 2010), that each hold information and together form a knowledge structure. In our case, the low-level visual features and rule-based associations are defined as schema "units" ("schema components" in our manuscript).

We agree with the reviewers that the multi-voxel patterns of low-level visual features likely constitute a representation of the circle pairs that were visually presented during retrieval trials. However, we argue that schemas entail associations between this low-level perceptual and higher-order information. During retrieval, visually presented circle pairs were combined with abstract, rule-based information, and could thus be used to predict the trial outcome. The combination of these different levels of information formed a simple schema.

Such a combination of perceptual content and higher-level information is also the case in the famous example of "living things" underlying semantic networks (McClelland et al., 1995). This network integrates low-level perceptual information ("it is yellow", "it has wings") with more abstract, higher-level information ("it is a canary", "it can grow"). Thus, although schemas are abstracted on the basis of multiple instances, we argue that at least a ‘summary representation’ of the perceptual features has to exist. We agree that "abstraction" of low-level visual features might be limited in our case. "Abstraction", however, happened through establishing simple associations between perceptual input and higher-order concepts (for example, that a circle on the left predicts “sun” when applying the spatial schema).

Indeed, as pointed out by Bartlett (1932) and as cited by Ghosh and Gilboa (2014), schemas are "abstracted" on the basis of multiple, variable encoding episodes. We demonstrated schema acquisition with increasing schema performance across day 1 (Results, Behavioral performance), which reflects the build-up of relationships between low-level visual features and rule-based associations. Schemas were learned quickly but did not depend on single-shot learning. Therefore, schema learning appeared to be based on multiple encoding episodes.

**Summary and action taken:** We thank the reviewers for this critical point. In our revised Introduction we now explain in detail why the low-level visual features constitute a schema component (or "unit"):

**Introduction:***‘’Crucially, our controlled design allowed us to independently capture the different schema components. During retrieval, visually presented circle pairs had to be combined with abstract rule-based information and could thus be used to predict specific trial outcomes. The combination of these different levels of information formed a simple schema. Therefore, the schema components consisted of (1) rule-based associations, and (2) low-level visual features of the task material (Figure 1—figure supplement 1).”*

*7) Is it believed that subjects are undergoing "schema retrieval" on every trial of the retrieval task (especially given the number of trials they've completed by the beginning of day 2), or that subjects are actively maintaining the relevant well-learned rule throughout each block? Neural data from the transfer test might be particularly useful in looking at retrieval of the learned schemas, as the schemas would theoretically need to be retrieved in order to integrate the new information into the pre-existing knowledge. However, the nature of the transfer task seems like it would entail rule-updating only for the non-spatial task (learning new colors), rather than the spatial task (the left/right rule still applies); this may explain why transfer was faster for the spatial task.*

Subjects might have maintained the relevant schema throughout the block (cueing schema retrieval before every block was a necessary design feature for our MVPA analysis). However, we assume that subjects were undergoing schema retrieval on every trial. Even though the number of retrieval trials on day 2 was very large, active retrieval was necessary to infer the correct schema outcome.

As the reviewers pointed out, the transfer test entailed schema transfer mainly for the non-spatial rather than the spatial schema. This explains faster updating of the spatial schema during the transfer (**Results, Transfer test: new schema encoding and retrieval**). Changing the color of the stimulus set allowed us to test schema transfer on novel material while matching the difficulty of old and new rule-based associations. For the spatial schema, a change in position would have lead to an increase in difficulty. In our additional analysis that compares behavior between day 1 and the transfer test, we thus only take the non-spatial schema condition into account (Point 9).

The transfer test was performed inside the MR scanner to keep the experimental context identical. While it was designed to obtain behavioral evidence for schema generalization, it was not suited for standard fMRI analyses (low number of trials; 8 spatial/non-spatial encoding trials per run, 8 spatial/non-spatial retrieval trials per run; 32 trials in total per run). Therefore, we reported only behavioral results in our original manuscript. Nevertheless, to test for retrieval of the learned schemas during the transfer test, we circumvented this with additional MVPA analyses:

If subjects performed schema retrieval on day 2, as well as during the transfer test, neural signatures should not differ. Thus, applying a classifier (spatial vs. non-spatial) trained on data from day 2 to neural data of the transfer test should yield representations of rule-based associations within AG (Figure 1—figure supplement 1). As for our original MVPA analysis, we used a moving searchlight (8mm radius) to extract local voxel patterns throughout the entire volume and trained a classifier on correct and high-confident retrieval trials of day 2 (spatial vs. non-spatial; for a detailed description and statistical analysis of searchlight maps please see **Materials and methods, Multi-voxel pattern analysis**). Critically, we applied this classifier to neural data of the transfer test. For this we used all trials of both runs (encoding and retrieval, irrespective of correctness or retrieval confidence), since the number of trials during the transfer test was low (see above). Effects were tested for significance using cluster-inference with a cluster-defining threshold of *P* < 0.005 and a cluster-probability of *P* < 0.05 family-wise error (FWE) corrected for multiple comparisons (critical cluster size = 172 voxels).

In line with our prediction, rule-based associations were represented within the left middle occipital gyrus and AG (Figure 15). Therefore, neural signatures of rule-based schema associations did not differ between day 2 and the transfer test. We interpret this as evidence that subjects performed similar cognitive operations during both study phases, possibly reflecting retrieval mechanisms.

Additionally, we discriminated the low-level features (circle pairs 1 and 2 vs. circle pairs 3 and 4) of the task material and showed that also during the transfer test, multi-voxel patterns were mainly represented within occipital regions (Figure 15). Also during the transfer test, both levels of information converged within the AG. The location of convergence corresponded well with our result of schema convergence during day 2 (Figure 15).

Author response image 4.Generalization of schema component representations from day 2 to the transfer test.Multi-voxel patterns of rule-based associations and low-level visual features were shared across study phases. Cut-outs of the horizontal slice are magnified to appreciate the overlap of schema components. Blue depicts the left AG cluster showing overlap of schema component during day 2. For display purposes, all maps were resliced to a voxel dimension of 0.5 mm isotropic and are shown at *P* < 0.001, uncorrected Table 5 (updated manuscript). L – left. This figure is incorporated in updated Figure 8.**DOI:**
http://dx.doi.org/10.7554/eLife.09668.026

**Action taken:** We thank the reviewers for this suggestion and include the additional MVPA analyses of the transfer test in our revised manuscript (see **Results, Transfer test: multi-voxel representations of schema components**; together with new Figure 8 and Table 5 in the manuscript).

**Results, Transfer test: multi-voxel representations of schema components:***“In our final analysis, we tested the convergence of schema components during the transfer test. […] Furthermore, this confirms our finding that the left AG recombines schema components after consolidation.”*

We also updated our Methods section:

**Materials and methods, Multi-voxel pattern analysis:***“The transfer test consisted of two runs […] and a cluster-probability of P<0.05 family-wise error (FWE) corrected for multiple comparisons.”*

*8) While the MVPA analyses revealed regions that could discriminate between rules and visual stimuli relative to chance (i.e., 50% for binary classification), permutation tests would be a more rigorous test of these hypotheses, given the generally high levels of false positivity when performing voxel-wise-t-tests, and the often non-gaussianity of the accuracy probability distributions, etc. (e.g., Stelzer et al., 2013). Further, for each MVPA analysis (rule-based, visual) it would be informative to look into the SVM weight maps to see which voxels are more responsive for spatial vs. non-spatial trials, etc. Based on the interpretation that information about both spatial and non-spatial rules (and both visual "categories", e.g., left vs. right) is converging in angular gyrus, it would be helpful to show that these AG voxels are not selective for one type of information.*

We address the reviewers points in two parts:

**A) Permutation tests for statistical thresholding of MVPA searchlight maps:** To compare the results of our MVPA with data obtained from random permutations, we closely followed the approach proposed by Stelzer and colleagues (Stelzer et al., 2013). Although theoretical chance level lies at 50% (binary discrimination), empirical discrimination values might exceed this threshold (false positives) due to the large number of independent tests, the low number of observations, and possible non-gaussanity of accuracy distributions (Golland and Fischl, 2003; Stelzer et al., 2013). The approach proposed by Stelzer and colleagues (2013) circumvents these problems by creating empirical null-distributions on a single-subject level (using non-parametric permutation tests). The null-distributions are then aggreated to a group level (using bootstrapping), resulting in a threshold map for voxel-wise chance-levels.

To this end, we repeated our searchlight discrimination for rule-based associations (spatial vs. non-spatial; see **Materials and methods, Multi-voxel pattern analysis** for a description of the analysis pipeline) for each subject (n = 23), but trials were shuffled randomly 100 times (i.e. with 100 permutation schemes, resulting in 2300 whole-brain permutation maps; Stelzer et al., 2013). To account for spatial correlations and the class label-trial relationship for a given permutation, the permutation scheme was kept constant for each whole-brain searchlight map. Maps were normalized using DARTEL and smoothed with a 3 mm Gaussian kernel (FWHM; as also in our original manuscript; Materials and methods, Multi-voxel pattern analysis).

Next, we aggregated single-subject permutation maps at a group level. We performed bootstrapping by randomly drawing one of the 100 permutation maps from each subject (with replacement), and averaging the 23 maps. This step was repeated 10^5^ times (Chen et al., 2011; Stelzer et al., 2013). We then created voxel-wise, empirical null-distributions based on the 10^5^ averaged permutation maps and determined chance-level (i.e. the accuracy that is observed for a random class label-trial relationship) by thresholding the normalized distribution at the 99.99% percentile (*P* < 0.001, thus accepting a 0.1% probability for a type I error, i.e. falsely rejecting the null-hypothesis of no difference between the classes). This yielded one whole-brain threshold map. The original searchlight map (with correct class label-trial relationships) was then compared to the permutation threshold map in a voxel-wise manner, only plotting voxels with a discrimination accuracy at or exceeding the permutation threshold.

Results confirmed our original MVPA analysis. We found discrimination performance of rule-based schema associations within the AG, significantly above the permutation threshold (Figure 16).

Author response image 5.Multi-voxel representations of schema components within left AG.The image shows magnified, horizontal cut-outs at the level of AG. Data represents significant discrimination of rule-based associations. Upper row: original MVPA results as reported in the manuscript (Results, Multi-voxel representations of schema components). Effects were tested for significance using cluster-inference with a cluster-defining threshold of *P* < 0.001 and a cluster-probability of *P* < 0.05 family-wise error (FWE) corrected for multiple comparisons (Table 3). Lower row: new MVPA results using the permutation framework. No cluster correction was applied. Both approaches yielded representations of rule-based associations within left AG.**DOI:**
http://dx.doi.org/10.7554/eLife.09668.027

**Summary and action taken:** We performed additional permutation tests that confirmed our original MVPA results. Despite parallel processing, this analysis required a large amount of computational resources (Single-subject permutations: 23 subjects*20 hours*100 permutations; Bootstrapping: 10^5^ bootstraps*2 hours; Voxel-wise distributions: 63 slices*35 hours). For this reason, and also because the permutation test confirmed our initial MVPA result, we did not complete this analysis for all other MVPA results reported in our original manuscript and retained our initial statistical threshold procedure. However, if the reviewers request this, we are willing to add this to our manuscript but then need to ask for a substantial extension to resubmit.

**B) SVM weight maps:** Linear Support Vector Machines (SVMs) allow the separation of, for example, two classes (binary discrimination). This is accomplished by the decision boundary surrounded by the maximum margins. The width of the margins is influenced by the data points closest to the decision boundary (the ‘support vectors’ that influence class discrimination the most). The final, optimal position of the decision boundary after training the model is described by the perpendicular weight vector (Bishop, 2007). The dot product of any data point with the weight vector thus tells us to which class the data point was assigned to (i.e. the SVM coefficients; positive for class +1, and negative for class -1), and can finally also tell us something about the importance of the specific data point (i.e. "feature") for class discrimination. Typically, this information is used by feature selection algorithms to improve discrimination performance (Guyon et al., 2002; Guyon and Elisseeff, 2003). Furthermore, ‘importance maps’ were presented previously (Polyn et al., 2005), however, they do not provide a measure of statistical significance of feature importance (Gaonkar and Davatzikos, 2013).

Most critically, SVM feature weights cannot be used to visualize which voxels are more responsive for spatial vs. non-spatial trials. For example, Haufe and colleagues (2014) demonstrated that feature weights should be interpreted analogous to "filters" that are needed to extract the signal with a high signal-to-noise ratio. Large feature weights can thus results from discriminating the two classes, but also from discriminating one class from noise. This is best explained by the authors themselves (Haufe et al., 2014): “We have shown that the parameters of multivariate backward/decoding models […] cannot be interpreted in terms of the brain activity of interest alone, because they depend on all noise components in the data, too. In the neuroimaging context, this implies that no neurophysiological conclusions may be drawn from the parameters of such models.”

**Summary and action taken:** We did not interpret the SVM feature weights due to the above-mentioned reasons and hope that we have convinced the reviewers with our explanation.

*9) For the behavioral analyses, it appears that the ANOVAs were conducted treating run as a categorical variable. While not critical for interpretation of the results, it would be more appropriate to treat run as a continuous variable, and test for a quadratic effect of run; simple effect analyses could then be used, e.g., to test for an effect of schema at run 1 if there is a significant run x schema interaction. Further, is it warranted to treat the retrieval runs on day 2 as from 1-7? Couldn't these be interpreted as runs 8-14? Finally, for transfer test analysis (it was unclear from the methods when the transfer test occurred, presumably on day 2; was this scanned?), if trying to show that subjects were applying knowledge learned on day 1 (and thus transfer from learning on day 1), effectively using the "schemas" to learn more quickly, the analyses should be comparing performance on the transfer task to the relative initial learning runs from day 1, rather than within the two runs of the transfer test.*

We thank the reviewers for these suggestions. We address the reviewers’ points in two parts:

**A) Factor ‘run’ as a continuous variable:** Our study design entailed a delay of 24 hours between sessions. Although retrieval performance, confidence, and reaction times did not differ between the end of day 1 and day 2 (Materials and methods, Schema consolidation), neural data provided evidence for increased PCC and MPFC involvement during schema retrieval after 24 hours. We interpreted this as network reorganization in line with assumptions from systems consolidation(Frankland and Bontempi, 2005). Due to this difference between days, we initially refrained from treating the factor ‘run’ as a continuous variable (runs 1 to 14).

To address the reviewers’ request, we repeated the behavioral analysis for retrieval performance across days, using a run (1 to 14) × schema (spatial, non-spatial) ANOVA for repeated measures. α was set to 0.05, Greenhouse-Geisser correction was applied when appropriate, and significant interaction effects were followed up by paired-sample *t*-tests. We found a significant run × schema interaction (*F*(4.8,82.3) = 3.5, *P =* 0.007; main effect of run: *F*(4.6,77.4) = 3.2, *P =* 0.014; no main effect of schema: *P =* 0.057; see below), caused by significantly better retrieval performance for the non-spatial schema during the first run of day 1 (*t*(21) = -3.2, *P =* 0.005; Figure 17).

The quadratic effect for the run × schema interaction just failed to reach significance (*P =* 0.058), probably due to faster learning of the non-spatial schema already within run 1 of day 1 (Figure 17). We thus repeated this analysis for the two conditions separately (i.e. two repeated measures ANOVAs for the spatial and non-spatial schema condition respectively, with the factor run (1 to 14)). As expected, we found a quadratic effect of run for the spatial schema condition (*P =* 0.002; main effect of run: *F*(3.95,67.07) = 4.468, *P* < 0.0005), while this was not the case for the non-spatial schema condition (*P =* 0.304; no main effect of run: *P =* 0.557).

Author response image 6.Schema retrieval across days.Data represents the % of correct responses. Error bars denote ± Standard Error of the Mean (SEM). * marks a significant (*P* < 0.05) difference between the schema conditions within the first run of day 1.**DOI:**
http://dx.doi.org/10.7554/eLife.09668.028

**Summary and action taken:** In conclusion, this alternative analysis led to the same results as reported in our original manuscript (**Results, Behavioral performance**). We still do not think that the factor ‘run’ should be treated as a continuous variable (1 to 14), but rather continuously within each day (runs 1 to 7), thereby acknowledging the 24 hour-delay between days. We therefore retained our original approach.

**B) Comparison of schema performance between day 1 and the transfer test:** The transfer test occurred at the end of day 2 inside the MR scanner. While it was designed to obtain behavioral evidence for schema generalization, it was not suited for standard, univariate fMRI analyses (low number of trials; 8 spatial/non-spatial encoding trials per run, 8 spatial/non-spatial retrieval trials per run; 32 trials in total per run; but see our reply to Point 7 for novel MVPA analysis).

The transfer test entailed schema transfer mainly for the non-spatial rather than the spatial schema. This explains faster updating of the spatial schema during the transfer (**Results, Transfer test: new schema encoding and retrieval**). Changing the color of the stimulus set allowed us to test schema transfer on novel material while matching the difficulty of old and new rule-based associations. For the spatial schema, a change in position would have lead to an increase in difficulty. In our additional analysis that compares behavior between day 1 and the transfer test, we thus only take the non-spatial schema condition into account. Thus, we compared non-spatial schema performance between day 1 (run 1) and the transfer test (run 1) using paired-sample *t*-tests.

Subjects responded significantly faster during the transfer test as compared to day 1 (*t*(21) = 5.66, *P* < 0.0005). Schema encoding performance did not differ between the study phases (*P =* 0.894; Figure 13). Similarly, subjects responded faster when retrieval non-spatial schema material during the transfer test (*t*(21) = 3.06, *P =* 0.006), but retrieval performance and confidence did not differ significantly (retrieval performance: *P =* 0.312; retrieval confidence: *P =* 0.244; Figure 13).

**Summary and action taken:** In summary, subjects responded faster during schema encoding and retrieval in the transfer test as compared to the initial run on day 1. We take this as indirect evidence that subjects applied schema knowledge acquired on day 1 to solve the transfer test at the end of day 2. Non-significant differences in encoding and retrieval performance as well as in retrieval confidence between the study phases are likely explained by the high performance and confidence already during run 1 on day 1.

We thank the reviewers for this suggestion. We clarified the occurrence of the transfer test at several points in the revised manuscript. The results of the transfer test (behavioral results and new MVPA analyses described in Point 7) were moved to the main **Results** section. We added the additional behavioral analysis reported above in the section **Results, Transfer test: initial schema acquisition vs. new learning**. Please find the edited text below.

**Results, Transfer test: new schema encoding and retrieval:***“Schemas provide knowledge structures that help new but related information to be integrated more rapidly (Tse et al., 2007; van Kesteren et al., 2014). […] This allowed us to match the difficulty between old and new non-spatial rule-based associations while a change in position would have lead to an increase in difficulty for the spatial schema condition.” (The remaining text of this section is identical to our original manuscript.)*

**Results, Transfer test: initial schema acquisition vs. new learning:***“To investigate whether non-spatial schema knowledge was transferred from initial schema acquisition to new learning we started out by comparing non-spatial schema performance, RTs, and retrieval confidence between the initial runs of day 1 and the transfer test. […] We take this as indirect evidence that subjects applied schema knowledge to solve novel but related material.”*

Minor comments [abridged]:

*1) The authors say: "response positions were balanced within each run". What does this mean? That each trial gave a different response-to-outcome mapping (i.e., changed S/R responses with regard to left and right hand across trials)?*

Left and right positions of S/R response options were switched randomly across trials to prevent fixed response-to-outcome mappings. We clarified this in our revised manuscript.

**Materials and methods, Procedure:***“To prevent fixed response-to-outcome mappings, response positions were randomly switched and subjects had to make a button press with their left or right index fingers.”*

*2) The description of transfer test is confusing: "The stimulus set was changed into circles with different colors while keeping the same pair-wise arrangement". What exactly does this mean? You mention the same colors after this (e.g., yellow/blue/red). Does this mean you change what color is associated with the location/color condition respectively so participants have to apply the same task to a new color/location combination?*

Stimulus material consisted of two different stimulus sets (yellow, blue, red; or green, orange, pink). While one set was used for schema encoding and retrieval across day 1 and 2, the other set was presented during the transfer session. The order of stimulus sets was balanced across subjects and they were asked to apply the same task to this new color combination. We edited this in our revised manuscript.

**Materials and methods, Material and task:***“Colored circles were matched for size and color intensity, and formed two different stimulus sets (yellow, blue, red; or green, orange, pink). While one set was used for schema encoding and retrieval across day 1 and 2, the other set was presented during the transfer test at the end of day 2 (Figure 1). The order of stimulus sets was balanced across subjects.”*

*3) In the subsection “Univariate activation analysis”, please clarify what you mean by "i.e. conjunction contrast". You are contrasting the average of two conditions vs. a further condition; you are not conducting a proper conjunction analysis (as in your MVPA analyses). To avoid confusion, we suggest you not call this a "conjunction contrast".*

We removed the term “conjunction contrast” from the text (**Materials and methods, Univariate activation analysis**).

*4) Why does the design require that both tasks rely on color, but only one on location (i.e., "sun when the yellow circle is on the left")? Although the other circle always coincides in location with the yellow circle, the task instructions are clear to pointing to a color. As such, it seems whereas the "color" task is purely related to "color" the "location" task relates to the location of a specific color. Was there a reason for not making the instructions purely location-based?*

We are sorry for the unclear description of our design. In fact, the spatial schema condition purely relies on the horizontal position of one circle. Indeed, the color (yellow) coincides also with the change in location, but is not predictive for the trial outcome. The non-spatial, and thus color-based schema condition is related to the color of one circle (one circle was always in the middle and was thus not predictive for the trial outcome). We clarified this at several points in the manuscript (see examples below). Further, we improved the description of our experimental design and task throughout the manuscript (for example, **Introduction and Materials and methods, Materials and task**; see below) and in Figure 1, Figure 1—figure supplement 1, and Figure 1—figure supplement 2 (see also our response to Point 2).

**Introduction:***“These schemas were incorporated into a modified, deterministic weather prediction task (Knowlton et al., 1994; Kumaran et al., 2009) in which subjects had learned that colored circle pairs predicted specific but fictive weather outcomes (“sun”, “rain”), depending on the location (spatial schema) or color (non-spatial schema) of one of the circles (Figure 1; for a detailed description please see Materials and methods, Material and task).”*

**Materials and methods, Material and task:***“Circle pairs could be solved with two different schemas regarding 1) the horizontal position of one circle (spatial schema; for example, “a circle on the left predicts sun”), or 2) the color of one circle (non-spatial schema; for example, “a blue circle predicts rain”).”*

*5) Why did you not orthogonalize the color/location pairs at retrieval (as you did at test), so you could then classify the "location" and "color" visual features separately? This seems like a stronger test of your AG effect – showing this region codes both task-related aspects.*

We designed our task to identify the overall representations of low-level visual features rather than the precise multi-voxel patterns that code for color and location. We apologize, but we do not see how the discrimination of color and location would constitute a stronger test of our AG effect. Visual input was kept constant between the spatial (location-based) and non-spatial (color-based) schema conditions. By showing that the AG holds representations of both schema conditions (and thus discriminates between rule-based associations), as well as low-level visual features, we show that this region codes for both task-related aspects.

*6) Why is it "unlikely that convergence of schema components emerges, for example, from spatial blurring between two neighboring, functionally distinct regions"?*

Reconsidering our statement, spatial blurring between two neighboring, functionally distinct regions cannot be excluded using our fMRI approach. In our revised manuscript we omitted this statement altogether since it collided with our revised structure.

*7) Retrieval performance was better for non-spatial than spatial on Day 1 – is this because of the rotation of items? The "rule" for the color (non-spatial) condition was the same as encoding with the modified stimuli.*

Indeed, better schema performance for the non-spatial relative to the spatial schema condition at the beginning of day 1 is likely due to the spatial rotation of the circle pairs between encoding and retrieval trials. Thus, while subjects could readily apply the non-spatial schema to retrieval trials, spatial schema knowledge required a transfer. However, performance for both conditions increased rapidly, which lead to near-ceiling performance already early during day 1 (**Results, Behavioral performance**).

*8) Is there evidence for consolidation over 24 hours? Would the same network of regions appear just as a function of run within a day, or does this only emerge after sleep? Similarly, if consolidation is "a prerequisite for mental schemas," then how were subjects able to perform so well on the retrieval task (arguably requiring some transfer from the encoding task) on day 1?*

Despite remaining insecurity about the time course of consolidation, recent literature has gathered evidence for consolidation processes taking place within 24 hours (van Kesteren et al., 2010). In this context, the role of the MPFC has received quite some attention (Frankland and Bontempi, 2005; Takashima et al., 2006). To clarify whether MPFC and PCC activation during retrieval is increased after a 24-hour-delay or rather increases as a function of run within a day, we used linear regression (as implemented in SPM8). Specifically, we tested for regions that would show a linear increase or decrease across runs (1 to 7) during schema retrieval (collapsing across spatial and non-spatial conditions; schema retrieval > perceptual baseline). Activation was tested for significance using cluster-inference with a cluster-defining threshold of *P* < 0.001 and a cluster-probability of *P* < 0.05 family-wise error (FWE) corrected for multiple comparisons (critical cluster size = 68 voxels).

During day 1 (Figure 18), we found increased activation within MPFC, PCC, as well as the left temporo-parietal junction, bordering the inferior AG. Conversely, the right AG, left supramarginal gyrus, the cerebellum, and bilateral fusiform gyrus showed a decrease in activation across the runs on day 1. During day 2, we did not find any significant activation increases or decreases during schema memory retrieval as a function of run.

Author response image 7.Changes in activation as a function of run on day 1.Increases in activation are shown in warm colors, decreases in cool colors. For display purposes, all maps were resliced to a voxel dimension of 0.5 mm isotropic and are shown at *P* < 0.001, uncorrected. Significant clusters are noted in Table 8. LH – left hemisphere; RH – right hemisphere; L – left.**DOI:**
http://dx.doi.org/10.7554/eLife.09668.029

Author response table 1.Changes in activation as a function of run on day 1. Retrieval (collapsed across spatial and non-spatial schema conditions) was compared to the perceptual baseline. Bold font indicates contrasts. MNI coordinates represent the location of peak voxels. We report the first two local maxima (> 8 mm apart) within each cluster. Effects were tested for significance using cluster-inference with a cluster-defining threshold of *P* < 0.001 and a cluster-probability of *P* < 0.05 family-wise error (FWE) corrected for multiple comparisons (critical cluster size = 68 voxels). L – left, R – right.**DOI:**
http://dx.doi.org/10.7554/eLife.09668.030MNIBrain regionxyzZ valueCluster sizeIncrease, run 1 to 7L cingulate gyrus-5-48204.3795L superior frontal gyrus-848354.33473R superior frontal gyrus560184.31R superior frontal gyrus155804.21L angular gyrus-52-62223.7468L angular gyrus-38-58283.57Decrease, run 1 to 7R middle occipital gyrus35-70304.69313R angular gyrus35-58453.87R superior parietal gyrus35-45423.85L supramarginal gyrus-42-40384.45122L superior parietal gyrus-45-45584.27cerebellum12-60-154.23301cerebellum0-48-104.22R fusiform gyrus32-40-183.92L inferior occipital gyrus-32-68-184.17199cerebellum-38-65-284.02

It is not surprising that we found partly overlapping activation maps, because one cannot assume entirely discrete retrieval processes for consolidated and unconsolidated memories and thus, these effects might partly be linked to task difficulty or performance. However, the consolidation contrast we reported in our manuscript equated runs for differences in retrieval performance, confidence, as well as reaction times (updated **Results, Schema consolidation**). Therefore, our result of increased MPFC and PCC activation after a 24-hour-delay is likely to reflect consolidation processes, whereas this might not be the case for activation changes throughout day 1.

We thank the reviewers for this comment. Despite some overlap with activity changes observed during day 1, we conclude that the increased MPFC and PCC involvement in the day 2 versus day 1 contrast reflects consolidation over 24 hours and thus retained this contrast to determine seeds for our connectivity analysis. To sum up, we think that overnight consolidation of schema memories took place.

*9) For the AG PPI analyses, how was the seed determined? Only the MPFC and PCC seeds are defined in the Materials and methods, Connectivity analyses. It looks like it’s just the local maxima pulled from the rule-based schema MVPA analyses, but is that region overlapping with the visual-features voxels? Also, it appears that there are some other similar sized regions of overlap between both MVPA analyses in the right hemisphere, more medial; would it be possible to get a table of all conjunctions?*

In our original manuscript, the AG seed for the reported PPI analysis was indeed determined by the local maximum of the rule-based schema MVPA analysis (Table 3). We placed a spherical seed with a radius of 8 mm around this seed coordinate, which overlapped with voxels that were found to be discriminative for the low-level features of the task.

However, reviewing our manuscript now, we understand that this might not have been the ideal choice for the AG seed region, because it does not optimally cover voxels that carry the representations of both rule-based associations and low-level visual features. We therefore created a new mask of the overlap between both MVPA analyses (rule-based schema, low-level visual features; 1437 voxels; updated Figure 6). Next, we repeated our PPI analysis using this cluster as a seed region (for details on PPI methods, please see **Materials and methods, Connectivity analysis**).

This yielded results similar to our previously reported PPI analysis (updated Figure 6): Spatial schema retrieval (compared to the perceptual baseline) was associated with increased functional coupling between the AG and its homologue in the right hemisphere, MPFC, PCC, inferior temporal gyrus, and bilateral fusiform gyrus (shown in red, critical cluster size = 88 voxels). A similar set of regions showed increased coupling with the AG during non-spatial schema retrieval (shown in blue, critical cluster size = 83 voxels). Connectivity profiles between the two conditions did not differ significantly (tested with a paired-sample *t*-test).

[Editors' note: further revisions were requested prior to acceptance, as described below.]

*The authors have responded well to the first round of reviews and have clarified/improved the manuscript. However, there is one issue that was not adequately addressed. This concerned showing the AG was not involved on Day 1. To answer this, they trained a classifier on Day 2, and applied this to the data on Day 1. Whilst interesting, this doesn't address my core concern. They need to train/test on Day 1 (i.e., exactly the same analysis they do for the Day 2 data). If this reveals the AG, this would clearly undermine their conclusion in relation to "consolidated" schema. The classification from Day 2 retrieval to the transfer task on Day 2 is interesting, and at least shows some degree of generalisation. However, there are many reasons why one might expect to see this effect, but not a Day 2 to Day 1 effect – not least because the latter is comparing across two completely separate scanning sessions.*

We ran the requested analysis to test the AG involvement during schema retrieval on day 1, mimicking the MVPA regime we previously used for the analysis of day 2 (**Materials and methods, Multi-voxel pattern analysis**). In brief, we trained and tested two classifiers for each local searchlight pattern using 7-fold cross-validation (rule-based associations: spatial vs. non-spatial; low-level visual features: circle pairs 1 and 2 vs. circle pairs 3 and 4; trials per category, mean ± s.d.: 54 ± 5). We included all retrieval trials in this analysis (irrespective of correctness or retrieval confidence), since day 1 contained a substantially lower amount of retrieval trials (8 trials per condition, per run).

We did not find significant representations of rule-based associations within the AG or any other brain region on day 1 (critical cluster size = 75 voxels). In contrast, low-level visual features were represented as expected within occipital regions, extending into the AG, as well as within the right anterior temporal lobe (Figure 19; critical cluster size = 80 voxels). These findings, namely representations of low-level visual features but not of rule-based associations on day 1, remained when we repeated the analysis in 14 subjects, selecting only correct retrieval trials with high confidence ratings (excluding nine subjects that showed one or more runs without correct and high confidence trials on day 1; trials per category, mean ± s.d.: rule-based associations, 48 ± 7 vs 50 ± 7, critical cluster size = 61 voxels; low-level visual features, 48 ± 8 vs 50 ± 6; critical cluster size = 65 voxels). To conclude, we found representations of low-level visual features but not of rule-based associations within the AG.

Author response image 8.Multi-voxel representations of schema components on day 1.(**A**) We did not find any significant representations of rule-based associations during retrieval on day 1. (**B**) Representations of low-level visual features were located within occipital regions, extending into the AG, as well as within the right anterior temporal lobe. For display purposes, all maps were resliced to a voxel dimension of 0.5 mm isotropic and are shown at *P* < 0.001, uncorrected. Significant clusters are noted in Table 6 (updated manuscript). L – left. This figure is incorporated in Figure 10.**DOI:**
http://dx.doi.org/10.7554/eLife.09668.031

These novel results corroborate our conclusion that the AG supports the recombination of schema components only after a 24-hour-delay. Here, we were able to identify representations of low-level visual features, but not of rule-based associations on day 1. Additionally, as we showed in our previous revision, schema components did only partly generalize between days. That is, while representations of low-level visual features were shared between days, representations of rule-based associations were not. We think that the coherence of these results suggests that our effects are not merely due to the separate fMRI sessions or the lower amount of retrieval trials during day 1, but due to a change in the underlying representations, in particular for the rule-based associations, that emerges after 24-hour-consolidation.

**Action taken:** We included the new analysis in our revised manuscript (**Materials and methods, Complementary analysis: AG involvement in schema retrieval on day 1**) and refer to it at several occasions in the **Results** and the **Discussion** sections. Please find the edited text passages below. Finally, we updated our analysis code and made it publicly available on GitHub: https://github.com/isabellawagner/searchlight-svm.

**Results, Multi-voxel patterns of schema components:***“However, this does not preclude the involvement of the AG in schema retrieval prior to 24-hour-consolidation, but may be caused by representational differences between the days. […] Complementary analysis: AG involvement in schema retrieval on day 1), suggesting that the left AG recombines schema components only after a 24-hour-delay.”*

**Discussion:***“Crucially, both schema components converged within the left AG on day 2 (Figure 5). […] This is corroborated by studies showing increased involvement of a parietal network in the processing of remote mnemonic content (for a review, see Gilmore et al., 2015).”*

Finally, we also adjusted our **Materials and methods**:

**Materials and methods, Complementary analysis: AG involvement in schema retrieval on day 1:***“We performed additional MVPA analyses to test the AG involvement during schema retrieval on day 1. […] Again, we did not find significant representations of rule-based associations, and low-level visual features were mostly represented in occipital regions (Table 6, lower part).”*